# Convergent evolution of Y chromosome gene content in flies

Shivani Mahajan[1] & Doris Bachtrog[1]

Sex-chromosomes have formed repeatedly across Diptera from ordinary autosomes, and X-chromosomes mostly conserve their ancestral genes. Y-chromosomes are characterized by abundant gene-loss and an accumulation of repetitive DNA, yet the nature of the gene repertoire of fly Y-chromosomes is largely unknown. Here we trace gene-content evolution of Y-chromosomes across 22 Diptera species, using a subtraction pipeline that infers Y genes from male and female genome, and transcriptome data. Few genes remain on old Y-chromosomes, but the number of inferred Y-genes varies substantially between species. Young Y-chromosomes still show clear evidence of their autosomal origins, but most genes on old Y-chromosomes are not simply remnants of genes originally present on the proto-sex-chromosome that escaped degeneration, but instead were recruited secondarily from autosomes. Despite almost no overlap in Y-linked gene content in different species with independently formed sex-chromosomes, we find that Y-linked genes have evolved convergent gene functions associated with testis expression. Thus, male-specific selection appears as a dominant force shaping gene-content evolution of Y-chromosomes across fly species.

---

[1] Department of Integrative Biology, University of California Berkeley, Berkeley, California 94720, USA. Correspondence and requests for materials should be addressed to D.B. (email: dbachtrog@berkeley.edu)

X and Y chromosomes are involved in sex determination in many species[1]. Sex chromosomes are derived from ordinary autosomes, yet old X and Y chromosomes contain a vastly different gene repertoire. In particular, X chromosomes often closely resemble the autosome from which they were derived, with only few changes to their gene content[2]. In contrast, Y chromosomes dramatically remodel their gene repertoire[3–5]. Y evolution is characterized by massive gene decay, with the vast majority of the genes originally present on the Y disappearing, and Y degeneration is often accompanied by the acquisition of repetitive DNA[4]. Old Y chromosomes typically contain only a few genes, and some lineages have lost their Y chromosome entirely[6]. The ultimate cause for Y degeneration is a lack of recombination on Y chromosomes, which renders natural selection inefficient[4]. However, while X chromosomes have been characterized and sequenced in many species, much less is known about Y gene content evolution beyond these very general patterns. Labor intensive sequencing of Y chromosomes in a few mammal species has revealed a surprisingly dynamic history of Y chromosomes, with palindromes retarding Y degeneration in primates[7], or meiotic conflicts driving gene acquisition on the mouse Y[8]. However, the repeat-rich nature of Y chromosomes has hampered their evolutionary studies in most organisms.

Dipteran flies have multiple independent originations of sex chromosomes[9]. In particular, flies typically have XY sex chromosomes and a conserved karyotype consisting of six chromosomal arms (five large rods and a small dot; termed Muller elements A-F[10]). Interestingly, we recently showed that superficially similar karyotypes conceal the true extent of sex chromosome variation in Diptera: whole-genome analysis in 37 fly species belonging to 22 families identified over a dozen different sex chromosome configurations in flies based on gene content conservation of the X chromosome[9]. The small dot chromosome was repeatedly used as a sex chromosome, but we detected species with undifferentiated sex chromosomes, others in which a different chromosome replaced the dot as a sex chromosome or in which multiple chromosomal elements became incorporated into the sex chromosomes, and others yet with female heterogamety (ZW sex chromosomes)[9].

However, no Y-linked genes were identified in our previous analysis, due to the difficulty in assembling genes from the often highly repeat-rich Y chromosome. Several Y-linked protein-coding genes in *Drosophila melanogaster*, for example, carry mega-base sized introns consisting of repetitive transposable element (TE) and satellite-derived DNA[11], making it impossible to assemble them using next-generation sequencing approaches[12, 13] (though the application of long-read PacBio

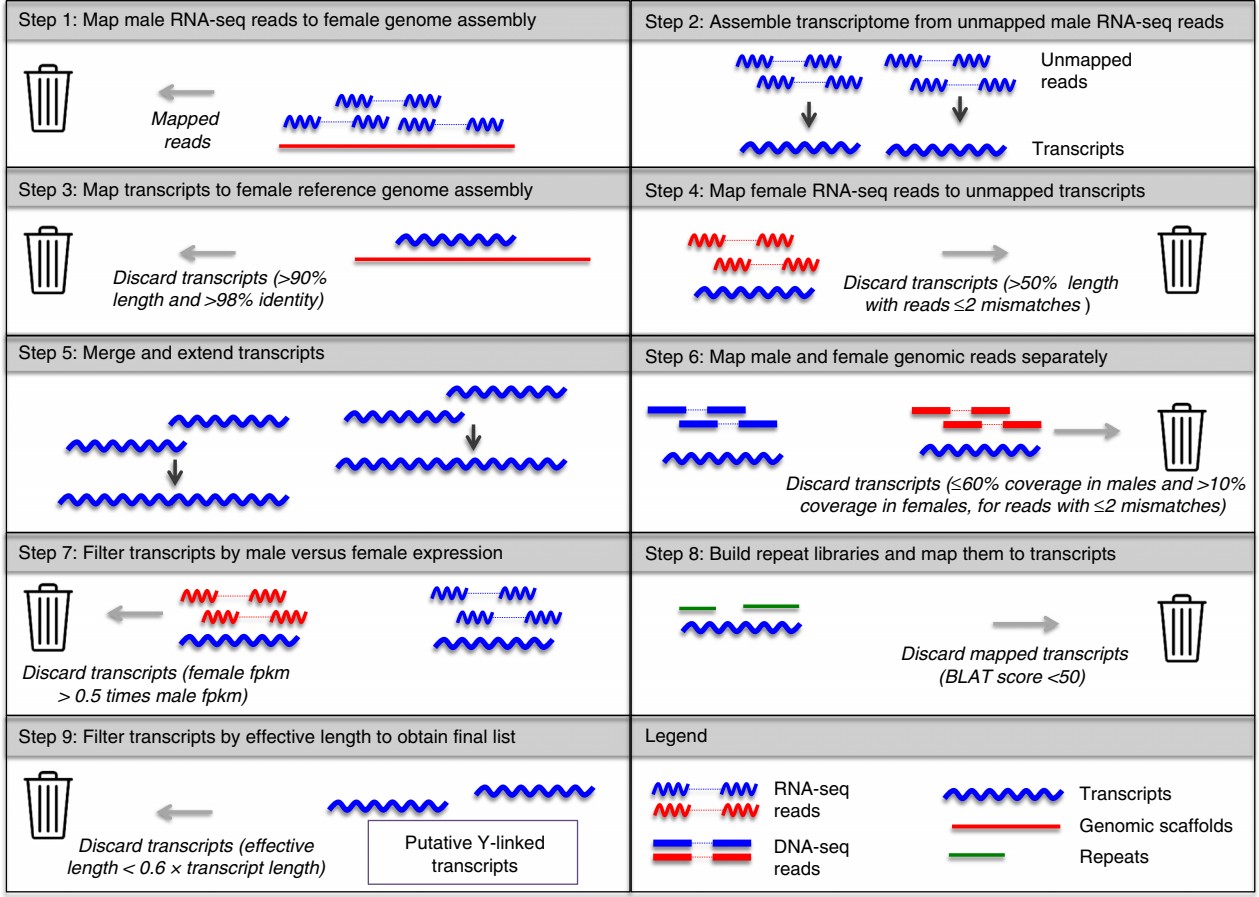

**Fig. 1** Bioinformatic subtraction pipeline to infer Y-linked transcripts. Male RNA-seq reads are mapped to genomic scaffolds build from female genomic reads (Step 1); unmapped male RNA-seq reads are used to build a de novo transcriptome (Step 2), and transcripts that either map to the female genome assembly (Step 3) or female RNA-seq reads (Step 4) are discarded. Remaining transcripts are merged (Step 5) and only merged transcripts are kept that show mapping to male genomic reads and no mapping to female genomic reads (Step 6) and that show expression in males but not females (Step 7). Transcripts that mapped to a de novo repeat library were discarded (Step 8), and only transcripts which had an effective length (as calculated by the software eXpress) greater than 0.6 times the transcript length were kept in the final list (Step 9)

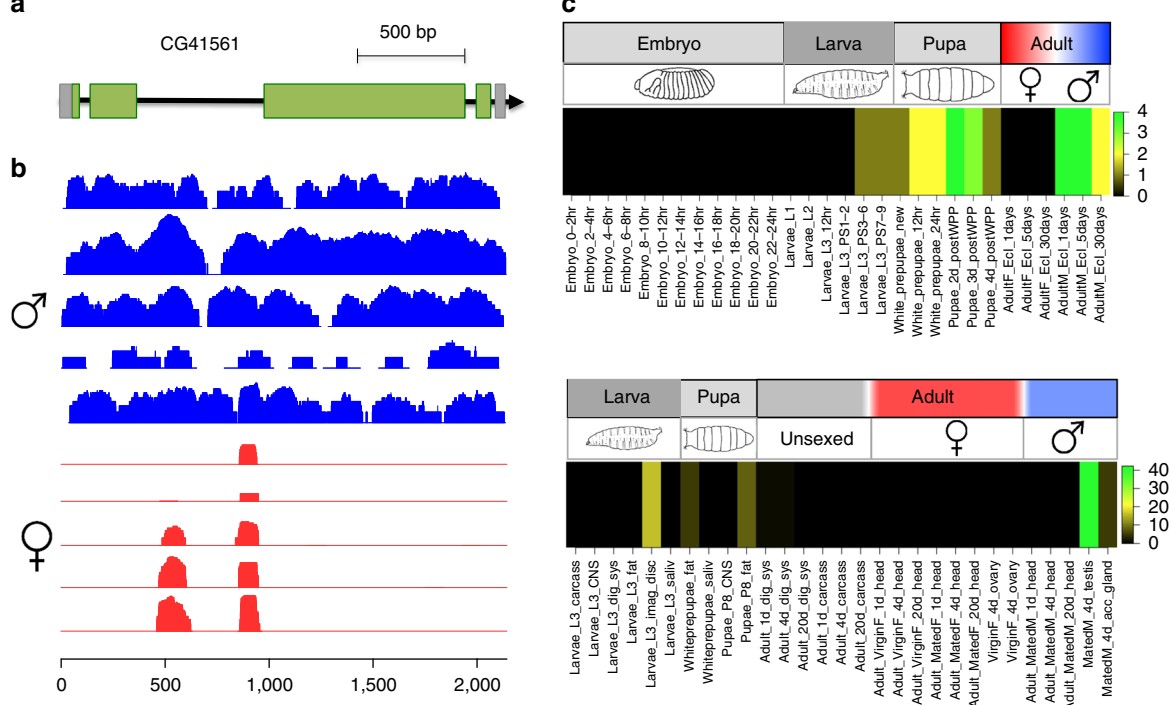

**Fig. 2** *CG41561*, a new protein-coding gene on the *D. melanogaster* Y chromosome. **a** Intron/exon structure of *CG41561* (*grey* are non-coding exons, *green* are coding exons). **b** Mapping of five male (*blue*) and five female (*red*) *Drosophila* genomic reads to *CG41561* (for strain information see Supplementary Table 2). **c** Expression profile of *CG41561* across developmental stages (*top*; samples are ordered by developmental time) and larval and adult tissues (*bottom*); colors in heatmap refer to expression level. *CG41561* is first expressed in third instar larvae, and shows maximum expression in pupae and young adult males (it is not expressed in females). Across tissues, *CG41561* is expressed in imaginal discs of third instar larvae and most highly in adult male testis. Expression profiles are taken from flybase. CNS, central nervous system; dig_sys, digestive system; fat, fat body; imag_disc, imaginal disc; saliv, salivary gland; acc_gland, accessory gland; 1d, 1-day; 4d, 4 days; 20d, 20 days

technology has proven useful in assembling Y-linked genes and genomic regions in *D. melanogaster*[14, 15]). Intriguingly, most Y-linked genes in *Drosophila* are not simply remnants of genes present on the autosome that became the sex chromosome; instead, they all appear to have been acquired secondarily on the Y, after it evolved its male-limited transmission[12, 13, 16, 17]. Y-linked genes in *D. melanogaster* all have male-specific functions and have adapted testis-specific expression, which suggests that they were acquired from autosomes and retained on the male-specific Y because of male-beneficial functions[12, 13, 16, 17]. This is in contrast to most mammalian species studied: while mammals have also acquired some multi-copy testis-specific genes secondarily, they still contain multiple genes that arose from genes ancestrally present on the proto-sex chromosomes with broad expression patterns and homologs on the X[7, 18–21]. These genes may have been maintained because of dosage constraints[20, 21].

Here we utilize whole-genome and transcriptome sequencing data from 22 Diptera species to trace gene content evolution of Y chromosomes in flies. Our sample encompasses sex chromosomes of very different ages, and at very different stages in their evolution. Our broad phylogenetic sampling across Diptera families focuses on old, independently formed Y chromosomes that presumably have been sex-linked for long time periods (i.e., several tens of millions of years), with basically no sequence homology left between the X and the Y[9]. *Drosophila* neo-sex chromosomes, on the other hand, were formed more recently (tens of thousands of years, to a few million years ago), by fusions of different autosomes to the ancestral sex chromosome pair of *Drosophila* (which is conserved across Drosophilidae). For recent fusions, the neo-X and neo-Y still contain considerable homology

between them, and the amount of sequence homology progressively declines for older fusions as Y chromosomes degenerate[4, 22–24]. This contrast enables us to infer the selective regime under which Y chromosomes evolve initially when still containing most of their ancestral genes, and their long-term evolutionary dynamics after most of their original genes have been lost.

In particular, our sampling scheme allows us to compare Y gene complement evolution on three different levels: (1) gene content evolution on old, non-homologous Y chromosomes across Diptera families; (2) the dynamics of gene gain and loss on the ancestral homologous Y chromosome of Drosophilidae; and (3) modification of the original gene complement on young, recently formed *Drosophila* neo-Y chromosomes. Here we identify Y-linked genes in 13 Diptera species, using a subtraction pipeline that infers Y genes from male and female genome and transcriptome data. We show that most Y genes on old Y chromosomes in flies are derived from autosomes, and have convergently evolved male-specific functions.

## Results

**Inference and validation of Y-linked genes in *D. melanogaster*.** Previous studies used male and female genomic data to identify Y-linked genes in *Drosophila* or *Anopheles* species[17, 25–27]. In particular, by comparing male and female sequence data to a reference genome, Y-linked sequences can be identified based on being present only in the male sequence data (either by identifying scaffolds with male-specific kmers[26] or by finding scaffolds with higher read coverage in male relative to female genomic reads[25]). Our initial application of these approaches to our male and female genomic fly data was of limited success to

 3

reliably identify Y genes[9], presumably due to a combination of factors: Y chromosomes have few genes and mainly consist of repetitive DNA, and our genome assemblies for the various fly taxa from next-generation sequencing data are more fragmented than the well-curated *Drosophila* or *Anopheles* genomes, and especially so at repeat-rich regions. Thus, fragmented genome assemblies combined with moderate genomic read coverage prevented us from using methods to infer Y-linked genes simply based on genomic data.

Instead, we developed a bioinformatics subtraction pipeline to identify Y-linked genes, using both transcriptome and genome assemblies and raw sequencing reads from both sexes (Fig. 1), which is similar to an approach performed in mammals[21]. Briefly, male transcripts were assembled from male RNA-seq reads that did not map to a female genome assembly, and Y identity was confirmed by mapping to male genomic and transcriptomic reads, and no/little mapping to female genomic and transcriptomic reads (Fig. 1, Methods).

We validated our pipeline by applying it to genomic and RNA-seq data that we collected for *D. melanogaster* males and females (Supplementary Table 1), and we could recover all previously identified Y genes, with the exception of the recently acquired *FDY* gene (Supplementary Fig. 1). *FDY* still shares considerable homology with its autosomal paralog (98% nucleotide identity[14]), and thus does not pass our strict bioinformatics filters. Our *D. melanogaster* assemblies of Y-linked transcripts are also highly contiguous and span almost all of the annotated coding sequences on the *D. melanogaster* Y chromosome (Supplementary Fig. 1). Most genes are covered by a single, full-length transcript, and four genes are covered by two partial transcripts with short gaps; only the *PRY* gene is missing a substantial fraction of its coding sequence in our *de novo* transcriptome assembly (the missing fragment did not pass the genomic coverage threshold in our pipeline). Moreover we were also able to recover genes from the *Mst77Y* gene family (Supplementary Fig. 1), which still retain moderate levels of homology to their autosomal paralog *Mst77F* (~90% identical at the protein level[28]).

In addition to the known Y genes, we identify one previously unmapped coding transcript on the *D. melanogaster* Y that corresponds to the annotated *CG41561* gene (which was suspected to be Y-linked[29]). This protein-coding gene is located on an unmapped 16.1-kb long scaffold, and has four annotated coding exons (Fig. 2a). We confirmed Y-linkage of that gene by read mapping to other published *D. melanogaster* male and female strains: *CG41561* was present in all males sequenced from various locations, but absent in reads derived from females (Supplementary Table 2, Fig. 2b). This supports our conclusion that *CG41561* is Y-linked in *D. melanogaster*, and fixed among *D. melanogaster* strains. Expression profiles show that *CG41561* is expressed predominantly in testis, and to some extent also in L3 larvae (Fig. 2c, Supplementary Fig. 2). We could not detect a paralog in the *D. melanogaster* genome for *CG41561* (even at low stringency), and orthologs were found within the melanogaster species group of Drosophila (Supplementary Fig. 3). Thus, like most other *D. melanogaster* Y genes[12, 13, 16, 17], *CG41561* does not have an old X homolog, and has a male (testis) function.

To infer the false-positive rate of our approach, we applied the same subtraction pipeline to identify female-specific transcripts by switching the sexes (i.e. assemble female transcripts that map to female genomic reads, but not to a male genome assembly or genomic reads, or male transcriptome data). We identify three putative female-specific transcripts, all of which are derived from the gene *kirre* that is located on the *D. melanogaster* X chromosome, and which shows higher expression in adult females compared to males. X-linked genes have reduced read coverage in males relative to females, and are thus more likely to be mis-inferred as female-specific. Overall, our pipeline shows both high sensitivity and specificity for detecting Y-linked genes, especially for species and genomic regions with high read coverage.

**Identification of Y genes across Diptera**. We initially applied our pipeline to 22 fly species for which we obtained genome and transcriptome data (Supplementary Table 1). Inferred genome sizes vary dramatically across the species investigated[9] (between 103 and 937 Mb; Supplementary Table 3). Overall, the quantity and quality of data collected is roughly comparable among species and similar to the *D. melanogaster* data analyzed above (between 8–87 million genomic reads per species, with more reads collected for species with larger genome sizes; Supplementary Table 3), suggesting that our power and sensitivity to detect Y-linked genes in other species should be roughly similar to that in *D. melanogaster*. However, genome size and the quality of genome and transcriptome assemblies, and to some extent, genomic read coverage, differ considerably among species. For instance, N50 for genome assemblies vary between 1 and 242 kb (Supplementary Table 3), and species with larger inferred genome sizes tend to have more fragmented genomes (Supplementary Table 3). Thus, given the sensitivity of our pipeline to genomic coverage, and genome/transcriptome assembly qualities, we applied our method to identify both male- and female-specific transcripts for each of the species, in order to empirically assess our false-positive rate. We failed to detect male-limited transcripts in four species: *Coboldia fuscipes* (the species with the smallest and most contiguous genome); the Hessian fly *Mayetiola destructor* (where males are known to lack a Y chromosome, i.e. they are X0); *Megaselia abdita* (a species with homomorphic sex chromosomes), and the flesh fly *Sarcophaga bullata* (which has a pair of small X and Y chromosomes). In three species, we find similar numbers of male- and female-limited transcripts: *Chironomus riparius* and *Aedes aegyptii* both have homomorphic sex chromosomes (and *A. aegyptii* has the largest inferred genome size of all species analyzed; Supplementary Table 3); and *Condylostylus patibulatus* (a species with XY sex chromosomes, and the third largest inferred genome, Supplementary Table 3). We only considered species further for which we had more than twice as many male-specific than female-specific transcripts (excluding *Tipula oleracea* and *Bactrocera oleae*), thus leaving us with 13 species to identify putative Y-linked genes (see Supplementary Table 3).

We additionally verified that our pipeline is reliable in identifying Y-linked sequences, using two different approaches. (1) We determined the location of candidate Y-genes in a subset of species with published high-quality genomes (*Anopheles gambiae* and *Drosophila* species) and (2) we used PCR to test for male-specific amplification of candidate Y-genes for a subset of non-Drosophila flies. Consistent with the high specificity of our pipeline that we observed in *D. melanogaster*, we generally find that our candidate Y transcripts either map to previously identified Y-linked scaffolds, or to unplaced scaffolds (which likely are derived from the Y chromosome). In particular, all three candidate Y-transcripts that we found in *A. gambiae* map to the previously identified Y-linked genes *YG1* and *YG2*[27]. Furthermore, 12 candidate Y-linked transcripts identified in *D. pseudoobscura* show highly similar sequences in the published genome (>95% of nucleotides mapping to over 50% of the transcript using blastn), and 11 of them map to unplaced scaffolds in the *D. pseudoobscura* genome. If we map putative Y-linked transcripts of its close relative *D. miranda* to the

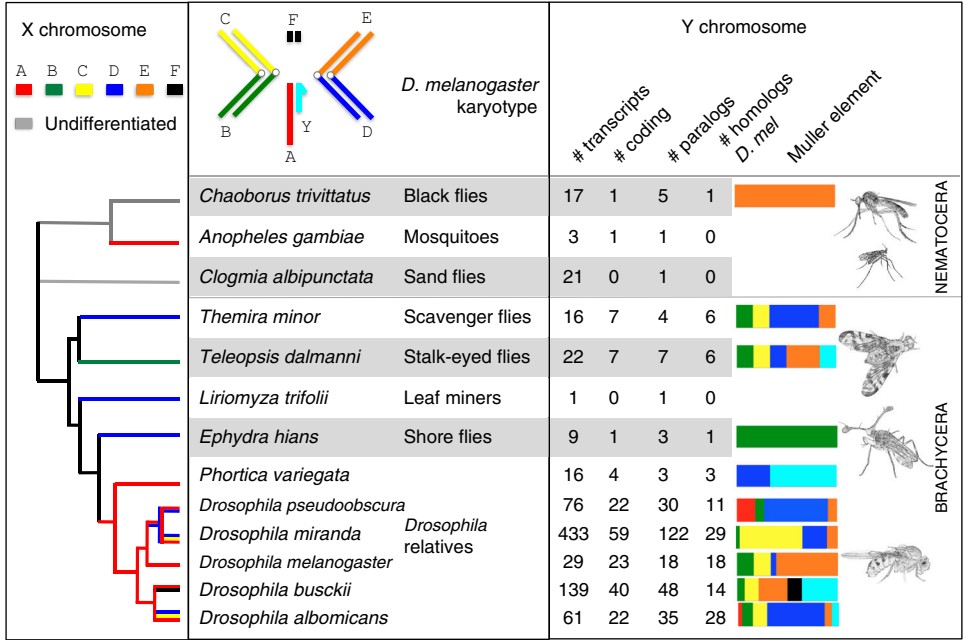

**Fig. 3** Y gene content evolution in flies. Shown are the species for which we have identified Y-linked transcripts. The karyotype of *D. melanogaster* males is shown (consisting of Muller element's A-F), and the color-coded branches of the phylogeny indicate which chromosome arm (Muller element) segregates as the sex chromosome in a species (from ref. [9]). The table gives the number of Y-linked transcripts identified from each species (#transcripts), the number of Y-linked transcripts that are predicted to be protein-coding (#coding), the number of Y-linked transcripts for which we could detect a paralog within the female genome of a particular species (#paralogs), and the number of Y-linked transcripts for which we could detect a homolog in *D. melanogaster* (#homologs *D. mel*). The bar charts indicate to which Muller element Y-linked homologs match within the *D. melanogaster* genome (with Y-linked genes of *D. melanogaster* being shown in turquoise); for *D. melanogaster*, mapping of Y-linked paralogs within the female *D. melanogaster* genome is shown. *Shading* distinguishes between different Diptera families (indicated by the common name of that family). Images drawn by Doris Bachtrog[9], modified and reproduced with permission

*D. pseudobscura* genome, we identify 63 transcripts that are highly similar to the reference genome sequence (> 95% of nucleotides mapping over 50% of the transcript); 20 of these transcripts map to unplaced (and thus putatively Y-linked scaffolds), 39 transcripts are located on Muller element C, which is the homolog of the recently formed neo-sex chromosomes in *D. miranda* (i.e., these transcripts are presumably derived from the *D. miranda* neo-Y chromosome), and only 4 map to other genomic locations. Thus, our pipeline is highly specific in each of the species in picking up true Y-linked sequences. PCR amplification in males but not females further confirmed Y-linkage for a subset of our putative Y-linked transcripts in several non-Drosophila species (6 transcripts in *Themira minor*; 10 transcripts in *Teleopsis dalmanni*; 4 transcripts in *Ephydra hians* and 8 transcripts in *Phortica variegata*, Supplementary Fig. 4; for transcripts and primers see Supplementary Table 4). To empirically assess a 'worst-case scenario' false-positive and false-negative rate, we subsampled our *D. melanogaster* data to match read counts with the species for which we have the lowest number of read pairs (*Mayetiola destructor*). Using this reduced dataset, we identify 25 male-specific and zero female-specific transcripts with our pipeline (we lose the *PRY* gene completely and fragments of a few other transcripts; see Supplementary Fig. 5). Thus, this further suggests that our approach is robust and sensitive to infer Y-linked transcripts across the species investigated.

**Fly gene repertoires**. In our study, we consider Y chromosomes at two very different stages of their evolution: old ancestral Y chromosomes from diverse Diptera families where most original Y genes have been lost[9], and young neo-sex

chromosomes of Drosophila, which may still contain most of their original genes[22–24, 30].

In total, we identified 187 protein-coding transcripts (or parts of transcripts), and 656 non-coding transcripts across all species that are potentially Y-linked. Note that the method that we use for classifying transcripts into coding vs. noncoding (i.e., Coding Potential Calculator[31]) is conservative in evaluating coding capacity of a DNA sequence, resulting in the assignment of a large number of transcripts as non-coding. Fragmented protein-coding transcripts, or short and highly divergent proteins (as is the case for many testis-expressed transcripts, see below) may be annotated as non-coding, and Coding Potential Calculator indeed called some incomplete Y-linked transcripts of *D. melanogaster* as non-coding, even though they mapped to parts of known protein-coding Y genes. The number of inferred Y genes varies substantially between species, with no protein-coding genes identified in *Clogmia albipunctata*, and 59 potentially protein-coding transcripts found in *D. miranda* (Fig. 3). We identify both Y-linked genes in species without morphologically distinguishable sex chromosomes, such as in black or sand flies, but also fail to detect Y genes in others with differentiated X and Y sex chromosomes (and high-quality genomes), such as in *Coboldia fuscipes* (Supplementary Table 3).

We previously showed that sex chromosomes originated independently in several fly families, and different chromosomes (termed Muller elements A–F; see Fig. 3) are segregating as the X chromosome in different species investigated[9]. Apart from the ancestral Y of Drosophilidae, all other sex chromosome systems investigated here evolved independently[9]. Drosophilidae are classified into two subfamilies, Drosophilinae and Steganinae, and we showed that the X chromosome of the two subfamilies is homologous (i.e. derived from Muller element A[9]). Two genes

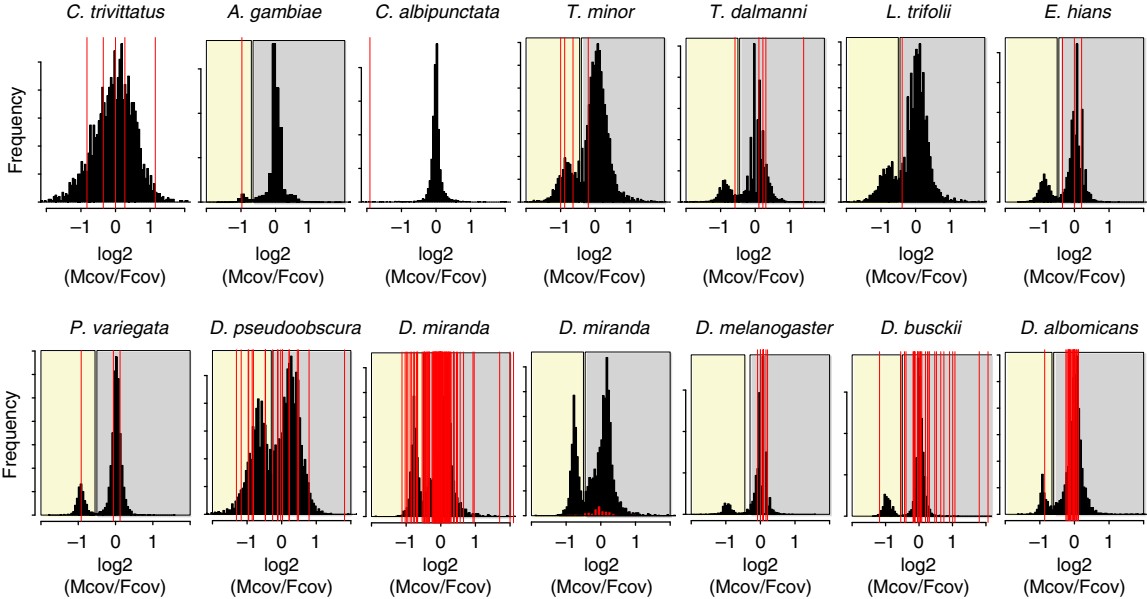

**Fig. 4** Genomic coverage of paralogs of putative Y-linked genes. The histograms show the male/female genomic coverage ratio of all scaffolds in the female genome (> 1000 bp). Scaffolds that are X-linked have reduced male/female coverage ratio (log2(Mcov/Fcov) around -1), while autosomal scaffolds have similar coverage in males and females (log2(Mcov/Fcov) around 0). The *red lines* indicate the male/female coverage ratio of scaffolds that have homologs on the Y chromosomes. Y-linked genes with X-linked homologs should show half the male/female coverage compared to ones with autosomal paralogs. Putative X-linked scaffolds are indicated by their *yellow shading*, and putative autosomal scaffolds are shown in *grey shading*. Note that we also show a histogram for paralogs of *D. miranda* Y-linked transcripts, due to the large number of Y-linked transcripts

that are Y-linked in *D. melanogaster*, *CCY* and *kl-2*, are also found in our list of putative Y-linked genes in *Phortica variegata*—a species from the Steganinae subfamily, indicating that they have been acquired on the Y chromosome in a common ancestor of both subfamilies. We also identified several Y-linked genes of *D. melanogaster* on the homologous Y chromosomes of *D. albomicans* (*kl-2* and *kl-3*) and *D. busckii* (*kl-2*, *kl-3*, *kl-5*, *ORY* and *PPr-Y*). As previously shown, the Y chromosome of flies from the *D. pseudoobscura* group is not homologous to the Y of *D. melanogaster*[30], and none of the ancestral Y genes in Drosophila are found among our putatively Y-linked genes in *D. pseudoobscura* and *D. miranda* (Fig. 3). Since sex chromosomes evolved independently in the other families of flies, we expect the gene content to differ among independently evolved Y chromosomes. Indeed, putative Y-linked genes identified in non-*Drosophila* species show no overlap; the only exception is *CCY*, which is found on both the Y chromosome of Drosophila (where Muller element A formed the sex chromosome pair), and also on the Y chromosome of the stalked-eyed fly *T. dalmanni* (where Muller element B formed the sex chromosome pair). Thus, this suggests that *CCY* was gained independently on both the Y chromosome of Drosophilidae, and the Y of stalk-eyed flies.

Y chromosomes may contain master sex determination genes, and in some cases, we could identify potentially interesting candidate genes for further study. In *Chaoborus trivittatus*, a species with homomorphic sex chromosomes, we were able to identify 17 potentially Y-linked transcripts, one of which is homologous to the DSX protein of several other Diptera species. The *dsx* gene is involved in sex determination in flies, and *dsx* homologs are expressed in the developing gonad of many animals, and have been utilized as master sex determination genes in both vertebrates and invertebrates[1]. We also recovered the *YG2* gene in *A. gambiae*, which is thought to be the male-determining gene in this species[27, 32].

**Origin of fly Y genes on ancestral Y chromosomes**. Y-linkage of genes could be a consequence of them being ancestrally located on the autosome that became a sex chromosome and escaping degeneration, or because genes were recruited to the Y chromosome secondarily (by translocations or transpositions) only after it became male-limited (as appears to be the case for most Y-linked genes in *D. melanogaster*[12, 13, 16, 17]. If current Y genes represent escapees of genes ancestrally located on the sex chromosomes, we expect that their closest paralogs in the genome map to the X. In contrast, if they secondarily moved onto the Y chromosome, we expect their closest paralogs to be autosomal. Note that we cannot distinguish genes that have been copied and moved to the Y from the X secondarily from those that were ancestrally located on the Y chromosome, based on location information alone (that is, we may overestimate the number of genes being ancestrally Y-linked).

We assessed the origin of our putative Y-linked genes in Diptera using two completely independent approaches. (1) We determined on which Muller element the closest homologs of Y-linked candidate genes in *D. melanogaster* are located. (2) We investigated whether the closest paralogs of putative Y genes within the same genome are X-linked or autosomal, based on genomic coverage analysis[9]. If current Y-genes are remnants of genes ancestrally present on the proto-Y chromosome, we expect them to map to the same Muller element (s) that formed the X chromosome in a species (using mapping information from *D. melanogaster*), and their closest paralog in the genome should be located on a genomic scaffold with half the male/female coverage ratio relative to autosomal ones (i.e., X-linked[9]). In contrast, if Y-genes were acquired from autosomes, we expect them to map to different Muller elements in *D. melanogaster* than the one(s) that formed the X chromosome, and their closest paralogs within a genome should harbor male/female genomic coverage ratios typical of autosomes. Note that mapping to *D. melanogaster* (i.e., our

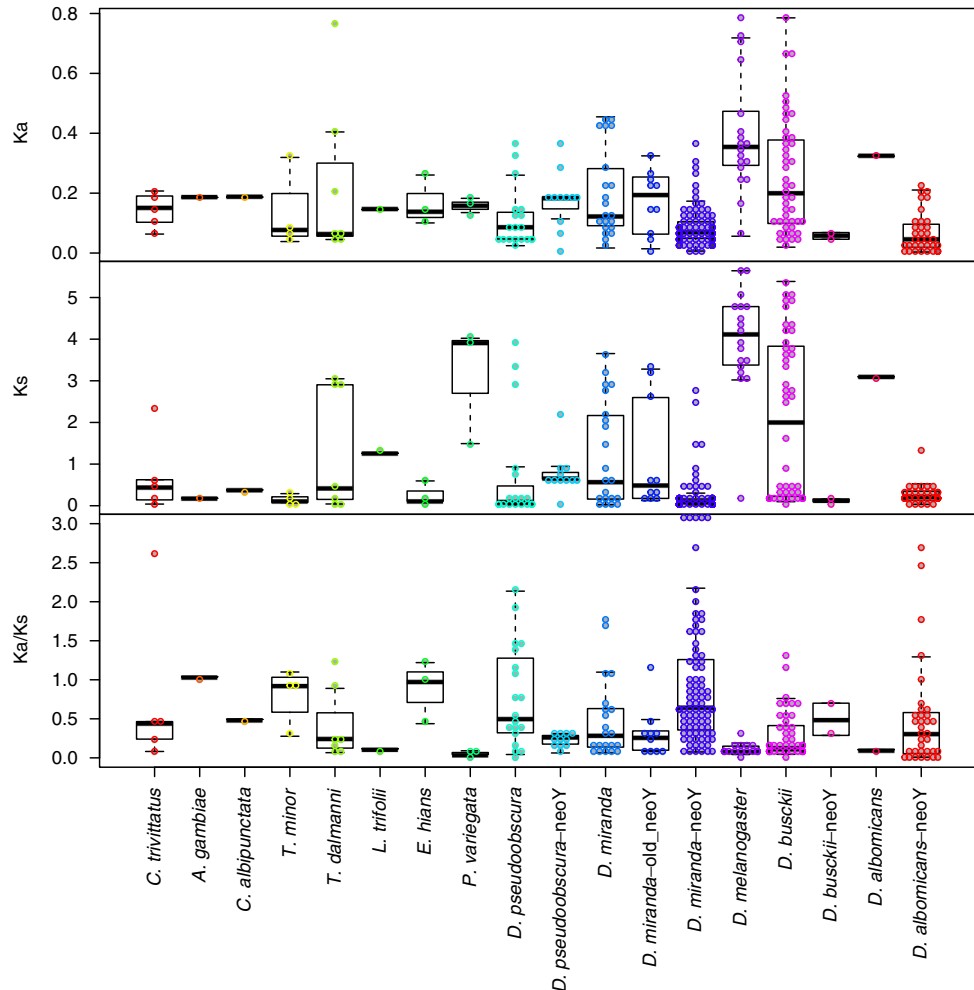

**Fig. 5** Divergence analysis of Y linked genes. Shown are rates of sequence evolution of Y-linked genes and their closest paralogs within the female genome. For *Drosophila* species with neo-sex chromosomes, we show divergence between neo-X/neo-Y homologs (i.e., Y-linked genes with their closest homolog on the neo-X) separately from other Y-linked genes. Shown are rates of amino acid evolution (Ka), rates of synonymous evolution (Ks), and their ratio (Ka/Ks). Note that Ka/Ks values > 3 are plotted at 3

first approach) assumes conservation in gene content of Muller elements across *Drosophila*, which has been found to largely hold true using comparative mapping[10] and whole-genome re-sequencing studies[33], and was validated by our previous comparative study inferring sex chromosomes across Diptera (where the vast majority of genes inferred as X-linked in various Diptera species, based on genomic coverage, mapped to a particular Muller element in *D. melanogaster*, and only few genes from other Muller elements, based on homology to *D. melanogaster*, were inferred to be X-linked based on coverage analysis[9]).

The suborder Nematocera is distantly related to fruit flies, and we detect only one homolog of a putative Y-linked gene in *Drosophila* (for a species with homomorphic sex chromosomes; see Fig. 3). We identify paralogs within the genome for three Nematocera species (two with homomorphic sex chromosomes, and one with heteromorphic sex chromosomes). The only paralog that we identify in a species with heteromorphic sex chromosomes (i.e. a transcript that partially overlaps with the YG1/YG2 genes in *A. gambiae*) is located on a scaffold with a male/female genomic coverage ratio typical of the X (Fig. 4), and mapping of this Y-linked transcript against the *A. gambiae* genome (https://www.vectorbase.org/) also confirms that its closest non-Y-linked paralog is located on the X chromosome

of *A. gambiae* (Supplementary Fig. 6; but note that the longer YG1 and YG2 transcripts mapped to autosomal locations in *A. gambiae*[27]). Indeed, a recent study utilizing a comprehensive RNA-seq dataset of sexed *A. gambiae* across development and whole and dissected adults (52 data sets in total) identified eight putative Y-linked genes (including the YG1 and YG2 genes), and found them all to be derived from autosomes[27]. It will be of interest to study additional Nematocera species with heteromorphic sex chromosomes, to better understand gene content evolution of the Y in this suborder.

Across most species belonging to the suborder Brachycera with old Y chromosomes (i.e., excluding *Drosophila* neo-sex chromosome systems), we find that putative Y-linked genes often have their homologs in *D. melanogaster* on several different Muller element's, and there is no overall enrichment for Y genes being derived from the same Muller element(s) that formed the X chromosome within a species (Fig. 3). Also, Y-linked genes have their closest paralog within a genome generally map to scaffolds that have male/female genomic coverage ratios typical of autosomes (Fig. 4). The Y chromosome of scavenger flies (*T. minor*), however, shows a somewhat different pattern: here, half of the identified putative Y-linked transcripts have their closest homolog map to the same Muller element that formed the X chromosome (3 out of 6; Fig. 3), and 3 out of 4 paralogs of

Y-linked transcripts show male/female coverage ratios in *T. minor* that are typical of X chromosomes (Fig. 4). Thus, a large fraction of Y-linked transcripts in scavenger flies may be remnants of genes initially present on the Y, while most putative Y-linked genes of stalk-eyed flies, shore flies, and Drosophilidae are derived from autosomes (consistent with *D. melanogaster* data[12, 13, 16, 17]; see Figs. 3 and 4). Hence, unlike in mammals, ancestral Y genes in flies are often derived from a wide variety of autosomal genes that were acquired on the Y chromosome only after it became male-limited.

Sequence divergence between putative Y genes and their autosomal paralogs allow us to roughly date when genes were acquired on the Y chromosome, with more recent acquisitions showing higher amounts of sequence similarity[12, 13, 16, 17]. We determined protein-coding paralogs for each putative Y-linked transcript in the female genome, and calculated rates of synonymous and non-synonymous substitutions between the Y-linked transcripts and their closest paralog in the female genome assembly (Fig. 5). In general, for non-*Drosophila* species, divergence between Y-linked genes and their autosomal paralogs is relatively low (Ka from 0.038 to 0.773 and Ks from 0.039 to 4.022), compared to divergence levels inferred in *D. melanogaster* (median Ka = 0.353, Ks = 4.112). Since we use *D. melanogaster* proteins to scaffold transcripts, the transcriptome assemblies for *Drosophila* species are more contiguous compared to the other species, which might make it more difficult to pick up more diverged paralogs in non-*Drosophila* flies. In general, we see a broad spread of divergence values between Y-linked genes and their paralogs, suggesting that genes were acquired on the Y chromosome at different evolutionary time points. This is consistent with patterns of gradual gene acquisition found on the *Drosophila* Y chromosome[17].

**Gene content evolution of *Drosophila* neo-sex chromosomes**. All *Drosophila* species investigated, apart from *D. melanogaster*, harbor neo-sex chromosomes. Here, fusions between the ancestral sex chromosome of *Drosophila* and an autosome incorporated a new chromosomal arm into the ancestral sex chromosome, at different evolutionary time points. The neo-sex chromosomes of the species we investigated form a temporal gradient and display various levels of degeneration. Unlike the ancestral Y chromosome of *Drosophila*, the gene repertoire of young neo-Y chromosomes still reflects their ancestral gene complement[22–24]. Our transcriptome analysis identifies some of the neo-Y genes as Y-linked transcripts, suggesting that they are sufficiently diverged at the DNA sequence level from their neo-X homologs to be identified by our bioinformatics pipeline. Indeed, for the species where a gene-rich autosome (i.e., not Muller element F) formed the neo-sex chromosomes, we generally see an overrepresentation of Y-linked genes derived from that Muller element that fused to the ancestral sex chromosome (Fig. 3). This suggests that they are remnants of genes originally present on the neo-Y.

The *D. albomicans* neo-X and neo-Y were only formed about 100,000 years ago, by the fusion of a large autosome consisting of Muller elements C and D to the ancestral sex chromosome, causing roughly 5000 genes to become sex-linked[24]. The neo-sex chromosomes of *D. albomicans* are still mostly homologous, with very little differentiation and degeneration of its neo-Y[24]. Our bioinformatics pipeline identified 61 Y-linked transcripts in *D. albomicans*. For 35 putative Y-linked transcripts for which we could identify paralogs in the female genome, 34 were added by the neo-Y fusion and one transcript is homologous to *kl-2* (we could not identify a paralog for the ancestrally Y-linked *kl-3* gene in the female genome of *D. albomicans*). Sequence

divergence between putative neo-Y genes and their neo-X homologs is much lower (median Ka = 0.04 and median Ks = 0.21) than divergence between ancestral *Drosophila* Y genes and their paralogs (Fig. 5), consistent with the recent formation of the neo-sex chromosomes in *D. albomicans*.

*Drosophila busckii*'s neo-sex chromosome system was formed by the fusion of the small dot chromosome (Muller element F, which contains only about 100 genes) to the ancestral sex chromosomes about 1MY ago, and it displays intermediate levels of Y degeneration[22]. Detailed molecular analysis suggested that the majority of neo-Y linked genes are still present, but about half appear pseudogenized[22]. Our bioinformatics pipeline identified 139 putatively Y-linked transcripts in *D. busckii*, and for 48 of those transcripts were we able to identify paralogous sequences in the female genome; two were added by the neo-Y fusion, 16 were ancestrally Y-linked, 21 autosomal, 2 from the ancestral X and 7 whose genomic location could not be determined based on mapping to their published genome[22], or homology with *D. melanogaster* coding sequences.

*Drosophila pseudoobscura* harbors an older neo-sex chromosome which arose about 15 MY ago (and which it shares with *D. miranda*). This system arose by the fusion of Muller element D (which contains roughly 3000 genes) to the ancestral X chromosome, and the fused arm is referred to as chromosome XR in the *pseudoobscura* group. Genes located on chromosome XR all appear hemizygous[23], and the evolutionary fate of the neo-Y of the *D. pseudoobscura* group has been unclear. Intriguingly, it has been shown that the ancestral Y of *Drosophila* became linked to an autosome in an ancestor of the *D. pseudoobscura* species group, at around the same time when the Muller element D–X chromosome fusion occurred[30, 34]. Consistent with this scenario, we do not detect any ancestral *Drosophila* Y genes as sex-linked in either *D. pseudoobscura* or *D. miranda* (Fig. 3). Since flies in the *pseudoobscura* group contain a morphologically distinguishable Y chromosome, it had been speculated that the current Y is the unfused neo-Y, i.e., the degenerated remnant of Muller element D[30]. Proof for this hypothesis, however, is lacking. Indeed, we find that many (13 out of 30) of the putative Y-linked transcripts that have mapped paralogs in the *D. pseudoobscura* genome are derived from chromosome XR (i.e., Muller element D). In addition, we identify 11 Y-linked genes in *D. pseudoobscura* that have homologs in *D. melanogaster*, and seven of them are located on Muller element D. This supports the idea that the current Y of *D. pseudoobscura* is derived from the unfused neo-Y. Interestingly, three of the seven Y genes that were ancestrally present on Muller D (i.e., also linked to Muller element D in *D. melanogaster*) have lost their former homologs on chromosome XR in *D. pseudoobscura*. Several studies have shown that X chromosomes in *Drosophila* are an unpreferred location for genes with male-specific function[35, 36], and all three genes that have been lost from XR are expressed predominantly in testis (both in *D. melanogaster* and *D. pseudoobscura*). Thus, 'demasculinization' of the X chromosome will further contribute to erode any remaining homology between the X and the Y, in addition to Y degeneration.

*Drosophila miranda* contains two neo-sex chromosomes that originated through independent fusions at different time points. It shares the ancient neo-X fusion with *D. pseudoobscura* (i.e., chromosome XR), and 25 different transcripts (corresponding to 6 genes) of the Y-linked transcripts in *D. pseudoobscura* are also Y-linked in *D. miranda* (20 of which are from Muller element D). Furthermore, *D. miranda* also harbors a more recently formed neo-sex chromosome: Muller element C became part of the ancestral Y chromosome only about 1.5 MY ago and has undergone massive degeneration, with over half of its genes

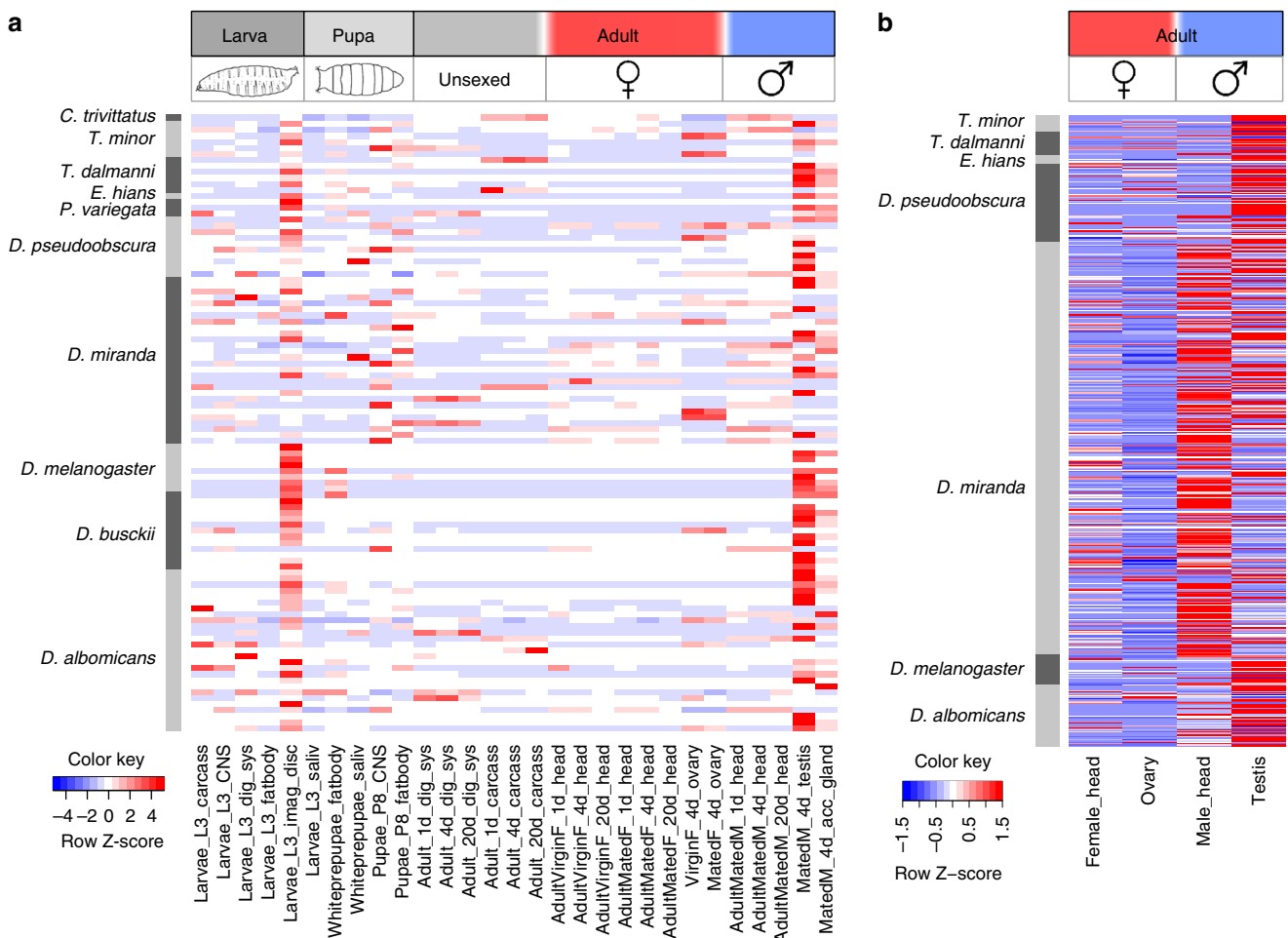

**Fig. 6** Functional specialization of Y-linked genes in flies. **a** Expression patterns of putative Y-linked genes with homologs in *D. melanogaster* in multiple *D. melanogaster* tissues. **b** Expression patterns of putative Y-linked genes in male and female head, and testis and ovaries for different species. Expression values were calculated as TPM (transcript per million) and row normalized to obtain *z*-scores with a mean of 0 and standard deviation of 1, using the built-in scale = 'row' argument in the heatmap.2 function from the package gplots in R

pseudogenized, and ~ 150 genes (of the roughly 3000 genes initially present on the more recently added neo-Y) have become deleted[23]. Consistent with its intermediate level of differentiation, we identify the largest number of Y-linked transcripts in *D. miranda*: there are still many genes left on the neo-Y, and neo-Y genes are diverged enough from their neo-X homologs to be detectable by our bioinformatics approach. We identified 122 transcripts with homologous sequences in the female genome, 10 of which are located on chromosome XR (and thus are supposedly from the 'ancestral' neo-Y fusion), 21 transcripts have been acquired from autosomes/the ancestral X of *Drosophila*, and 91 transcripts whose closest paralog is located on the neo-X. Also, the majority of genes with homologs in *D. melanogaster* map to Muller element C (18 out of 29). Again, sequence divergence for the young neo-Y genes (median Ka = 0.069 and Ks = 0.094) is lower than for ancestral Y genes or genes from the more ancient neo-Y that derived from the fusion of chromosome XR to the ancestral X (median Ka = 0.192 and Ks = 0.493; see Fig. 5).

**Functional evolution of Y genes.** Previous work[17, 27] and our analysis suggests that many genes on ancestral Y chromosomes were acquired from autosomal locations. The majority of genes on more recently formed neo-Y chromosomes, in contrast, eventually undergo massive degeneration, while some start to

diverge early on to be identifiable as male-specific in our pipeline (such as those on the *D. albomicans* or *D. miranda* neo-Y), or are maintained over long periods (such as on the *D. pseudoobscura* neo-Y). To assess which functional pressures are driving the acquisition of new Y genes, or the maintenance or divergence of existing neo-Y genes across flies, we used tissue-specific expression data. On one hand, we assessed expression of putative Y-linked genes with homologs in *D. melanogaster* (Fig. 3) in multiple *D. melanogaster* tissues. We find that most genes that have maintained or acquired Y-linkage are highly expressed in male-specific tissues of *D. melanogaster*, i.e., most genes are highly expressed in testis, and many are also highly expressed in male accessory glands (Fig. 6a). To test whether this enrichment for testis- or accessory gland-biased expression is significant, we calculated expression (as TPM; transcripts per million) for all annotated *D. melanogaster* genes (version 6.02) in 5 samples (male head, female head, ovary, testis, and accessory glands) and performed binomial tests to evaluate if genes that are Y-linked across Diptera are overrepresented for genes showing highest expression in testis or accessory glands relative to all annotated *D. melanogaster* genes (61 genes out of 106 in our Y-linked gene set vs. 5216 genes out of 17560 genes total; *P* < 0.0001) or whether they are expressed exclusively in testis and accessory glands (36 genes out of 106 in our Y-linked gene set vs. 1655 genes out of 17560 genes total; p < 0.0001). A subset of our putative Y-linked genes across Diptera have clear roles in spermatogenesis in

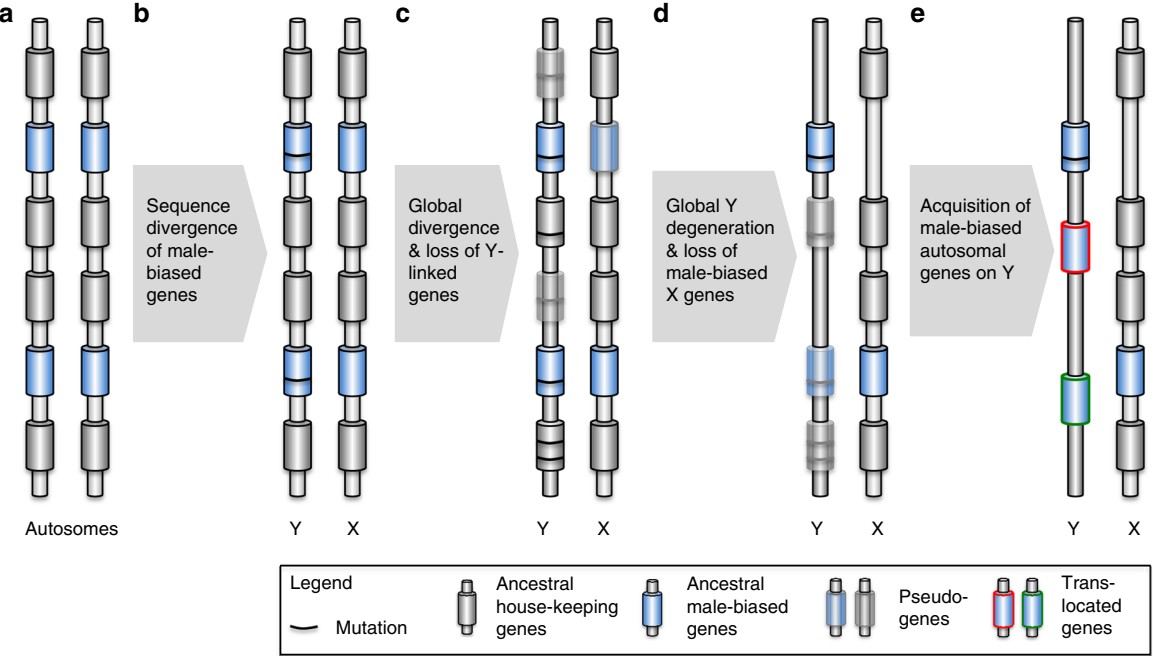

**Fig. 7** Model of Y-linked gene content evolution in flies. The dynamics of gene content evolution in flies across time is shown. **a** X and Y chromosomes originate from ordinary autosomes with identical gene content. **b** The first genes to diverge at the DNA sequence level are genes with male-biased expression. **c** Over time, most genes on the Y acquire mutations, and many start to become pseudogenes. **d** Continuing Y degeneration, and loss of some male-biased genes on the X chromosome. **e** Acquisition of male-biased genes from autosomes to the Y chromosome. Genes with male-biased expression are shown by *blue shading*, and genes with broad functions are shown with *grey shading*

*D. melanogaster*. In *Ephydra hians*, for example, a homolog of the *male sterile (2) 34Fe* gene is found on the Y chromosome, which is highly expressed in male testis, and involved in spermatid differentiation[37]; in *T. minor*, a homolog of the *Rcd7* gene is found on the Y, which is involved in spermatogenesis[37]; or the *yuri* gene on the Y of *D. miranda* and *D. pseudoobscura*, which is involved in sperm individualization[38]. All these observations are consistent with Y chromosomes being a preferred genomic location for genes with male-specific function[35, 36].

For a subset of species (*D. melanogaster*, *D. albomicans*, *D. miranda*, *D. pseudoobscura*, *E. hians*, *T. dalmanni*, *T. minor*) we had expression data from male and female head, as well as ovary and testis (Supplementary Table 5). This allowed us to compare tissue-specific expression patterns of Y-linked genes directly within a species. Again, we find that most Y genes show highest expression in testis compared to somatic tissue (Fig. 6b). This directly demonstrates that surviving or newly acquired Y genes are selected for their male-specific functions. Note that Y-linked genes may show male-specific expression either because their male-specific function makes the (male-limited) Y chromosome an ideal genomic location or because genes on the Y chromosome evolve male-specific functions in response to being located on the Y, and both processes have been found to be important in shaping the gene content of the human Y chromosome[18, 39]. The maintenance of testis-expressed genes on degenerating neo-Y chromosomes[23, 40] and the recruitment of genes to the Y chromosomes whose autosomal paralogs have ancestrally testis-biased expression (as for example inferred from expression patterns in *D. melanogaster*; see Fig. 6a or based on testis-biased expression patterns of autosomal or X-linked paralogs of testis-expressed Y-linked transcripts in *D. pseudoobscura*; see Supplementary Fig. 7) provides evidence that genes with male-biased expression are selectively acquired or preserved on the Y because of their benefit to males. However, it is possible that some Y-linked genes evolved male-specific expression in response to being located on the male-limited Y chromosome.

**Temporal evolution of Y chromosomes**. Comparison of tissue-specific expression patterns of *Drosophila* neo-Y chromosomes reveals an interesting temporal dynamics of Y gene evolution (Fig. 6b). On the very recently formed neo-Y chromosome of *D. albomicans*, the majority of genes are not yet differentiated sufficiently to be identified as neo-Y-linked by our pipeline, yet the subset of genes that have accumulated enough mutations so we can pick them up as being located on the Y are predominantly expressed in testis. In *D. miranda*, many more genes on the neo-Y have diverged sufficiently at the DNA sequence level from their neo-X homologs to be identifiable as male-specific, and while many are indeed highly expressed in testis, most are also expressed in somatic (head) tissue. On the older neo-Y of *D. pseudoobscura*, on the other hand, only few genes remain, yet those that have survived are predominantly expressed in testis. This temporal comparison of Y chromosomes paints a dynamic picture of Y gene content evolution, and reveals the importance of male-specific selection shaping Y differentiation (Fig. 7). At the earliest stages of Y chromosome formation (as in *D. albomicans*), the majority of genes are indistinguishable on the formerly identical sex chromosomes, and the first genes to diverge at the DNA sequence level on the Y are genes with male-specific function. As time progresses, most genes, independent of their function, start to differentiate and begin to degenerate on the non-recombining Y (as in *D. miranda*). On old Y chromosomes, almost all of the original genes have been lost, and only those with male-specific function will survive on the Y (as in *D. pseudoobscura*), or will be gained secondarily from autosomal paralogs (as in *D. melanogaster*).

**Loss of homology between diverging sex chromosomes.**
The lack of homology between the *D. melanogaster* X and Y chromosome has fueled speculation that the Y in this species is not a degenerate homolog of the X, but instead that the ancestral sex determination system of *Drosophila* was X0, and that the Y was acquired secondarily from a B chromosome[16]. Here, we show that X and Y chromosomes with little homology have evolved independently multiple times in Diptera, and three processes contribute to a lack of homology between X and Y chromosomes (Fig. 7). Massive gene loss on the Y is the dominant force shaping sex chromosome divergence, and 100 s of genes can quickly erode on a degenerating Y within a few million years. The few genes that are retained on the Y typically have male-specific function, yet exactly those genes are more likely to be lost from the X. In particular, female-biased transmission or the peculiar regulatory mechanisms of the X during spermatogenesis (such as transcriptional suppression of X-linked genes or a lack of dosage compensation in male germline[9, 41]; note that the causes of reduced expression of the X chromosome during spermatogenesis are controversial[42]) may make it an un-preferred location for testis-expressed genes, and demasculinization (i.e., loss of testis genes) may cause loss of genes on the X that are preferentially maintained on the Y. Finally, recruitment of autosomal genes (typically with male-specific expression) to the Y chromosome means that the closest homologs of many Y genes are located on autosomes.

Thus, our demonstration that Y chromosomes quickly lose homology with the X independently in many lineages with independently formed sex chromosomes and instead acquire genes of autosomal origin argues against the hypothesis that the Y of *D. melanogaster* derives from a supernumerary B chromosome. Furthermore, our comparative analysis in *Drosophila* demonstrates the gradual nature of loss of homology and the various mechanisms contributing to it, and there is thus no need to invoke any additional mechanism (such as a complete loss of the ancestral Y followed by the secondary recruitment of a "B" chromosome) to explain the observed lack of homology between the X and Y of *Drosophila*.

A prominent gene on the Y chromosome in *D. melanogaster*, and in fact the only locus that is shared between the X and Y, is the tandemly repeated rDNA gene family[43]. While there seems to be a general tendency for the rDNA locus to reside on the sex chromosomes in Diptera[44–46], in several species the rDNA is additionally or even exclusively located on autosomes[47–49]. The X and Y rDNA units in *D. melanogaster* are highly similar in sequence due to occasional exchange events[50], and can thus not be detected with our bioinformatics approach that identifies male-specific sequences.

## Discussion

The nature of Y chromosomes has remained mysterious. Here, we investigated the gene complement of Y chromosomes in flies, at very different stages of their evolutionary transition. Young neo-Y chromosomes allow us to study gene loss on gene-rich, degenerating Y chromosomes, and the selective forces driving the divergence and maintenance of a subset of genes that were originally present on the Y[4, 22–24]. Comparisons of the ancestral Y chromosome of *Drosophila* species enable us to investigate the dynamics of gene gain and loss on old, homologous Y's[12, 13, 16, 17]. Finally, the contrast of old, non-homologous Y chromosomes across Diptera families allows us to identify convergent evolutionary pressures operating on old Y chromosomes.

We find that male-specific selection is a dominating force shaping gene content at each stage of Y evolution. Testis-expressed genes are the first to diverge on very recently formed neo-Y chromosomes (such as in *D. albomicans*), and are preferentially retained during the initial period of massive gene loss on young, degenerating Y chromosomes (such as in *D. miranda* and *D. pseudoobscura*). Once the majority of genes has been lost, Y chromosomes continually reshape their gene complement, by constant losses and gains of genes derived from other locations in the genome with male-specific function[17]. Additionally, genes ancestrally present on the sex chromosomes with male function may be retained on the Y but lost on the X (as is the case for *D. pseudoobscura*). Thus, after long evolutionary time periods, all homology between the X and Y may be lost. While the 1.5MY old neo-Y of *D. miranda* still shows substantial homology with its former homolog, almost all traces of their shared ancestry have already eroded after 15MY of evolution for the *D. pseudoobscura* Y, and no homology remains between the ancestral sex chromosomes of *Drosophila*[12, 13, 16, 17].

Independently formed ancient Y chromosomes across flies have evolved similar characteristics convergently: they typically contain very few genes with male-specific function, which appear to be derived mainly from other autosomal locations instead of being remnants of genes ancestrally present on the Y[12, 13, 16, 17]. The long-term dynamics of ancestral mammalian Y chromosomes is somewhat different. Here, the Y has retained some of its ancestral genes, and they appear to have been maintained for ancestral gene dosage[20, 21]. Differences in the mechanism of dosage compensation may contribute to this difference: while male flies generally seem to restore the ancestral gene dosage of X-linked genes through hyper-expression of the X chromosome[9], male mammals appear not to globally upregulate X-linked genes[51, 52]. Thus, there may be stronger selection to maintain dosage-sensitive genes on the mammalian X chromosome. Therefore, both lineage-specific as well as general evolutionary mechanisms shape the gene content of Y chromosomes across species.

## Methods

**Data.** We utilized previously published data from separately sequenced male and female genomes for each of the 22 species in our study[9]. We also sequenced the transcriptomes from male and female whole body separately for each of those species, as described[9]. We obtained RNA-seq data for male and female heads as well as testes for *Drosophila albomicans, D. pseudoobscura, D. miranda, Ephydra hians* and *Themira minor*. Data for the same tissues and male and female whole body for *D. melanogaster* was downloaded from NCBI. Newly collected data have all been uploaded to GenBank. Supplementary Table 2 gives an overview of all the data sets used, including accession numbers for newly collected sequences.

Coding sequences and protein sequences for *Drosophila melanogaster* genome assembly version r6.2 were downloaded from flybase.org.

**Genome assembly.** For each species, male and female paired end genomic reads were trimmed and assembled separately using SOAPdenovo[53] with a kmer size of 31. An overview of the resulting genome assemblies is given in Supplementary Table 3.

**Transcriptome assembly.** FastQC (http://www.bioinformatics.babraham.ac.uk/projects/fastqc/) was used to quality filter the reads. After trimming, Trinity[54] was used to assemble the transcriptomes for each species using default parameters and a kmer size of 25. An overview of the resulting transcriptome assemblies is given in Supplementary Table 3.

**Pipeline to identify Y-linked coding sequences.** We used a subtraction approach to identify putative Y-linked sequences, similar to previous studies done in mammals[20, 21]; (Supplementary Fig. 1). Our pipeline starts by making a putative transcriptome assembly of Y-linked genes, and at each step filtering out possible false positives to obtain a conservative list of candidate Y-linked genes. In particular, we first map male RNA-seq reads to the female *de novo* genome assembly using tophat2[55] and then build a male *de novo* transcriptome assembly using Trinity[54] from RNA-seq reads that do not map to a female assembly using default parameters and a kmer size of 25. We then mapped the assembled transcripts to the female reference genome using BLAT[56] and discarded transcripts if greater than 90% of their length aligned with 98% or greater identity to the

female genomic scaffolds or if the blat score was less than 50. Following this, female RNA-seq reads were mapped to the remaining transcripts using bowtie2[57] with default parameters and transcripts that mapped 50% or more of their sequence with up to two mismatches were discarded. We then did a merging step using the software TGICL[58] using a minimum overlap of 30 bp and STM, i.e. scaffolding by translational mapping[59] to remove redundancy and obtain maximal length transcripts. We validated the merged transcripts by mapping them to genomic reads using bowtie2[57] with default parameters and allowing up to two mismatches. We used soapcoverage (http://soap.genomics.org.cn/soapaligner.html) to calculate male and female genomic coverage for each transcript, and only transcripts for which greater than 60% of their sequence was covered by male reads and less than 10% by female reads were retained. We then mapped male and female RNA-seq reads separately to the remaining transcripts using bowtie2[57] with default parameters and calculated RPKM values using the software eXpress[60]. Only transcripts with greater than twice the expression in males compared to females were retained. In order to eliminate transcripts with repetitive sequences, we built repeat libraries for each species using RepARK[61] and discarded transcripts that mapped to repeats using the software BLAT with default parameters. We then did a final filtering step and discarded transcripts if their effective length used to calculate RPKM as determined by eXpress[60] was < 60% of the total transcript length. We repeated the exact same pipeline but switching sexes in order to identify female-specific transcripts, to empirically assess the false-positive rate of our approach (Supplementary Table 3), and only kept species for further analysis where we identified at least twice as many male- relative to female-specific transcripts (Supplementary Table 3). Sequences of all assembled putative Y-linked transcripts are given in Supplementary Data 1.

**PCR validation for a subset on Y-linked genes**. DNA was isolated from two single male and female flies using the Qiagen DNeasy Blood/Tissue kit. PCR primers were designed using the Primer3 software based on assembled putative Y-linked transcripts. PCR amplification was performed with the ThermoFischer Scientific DreamTaq kit, with annealing temperatures ranging from 55 to 60 °C.

**Finding paralogs and determining Ks values**. We calculated Ka, Ks and Ka/Ks values for all Y-linked transcripts for which we could find paralogous sequences in the female genome assembly. To this end, we first determined putative peptide sequences for the candidate Y-linked transcripts using either CPC[31] or ORF finder (http://www.bioinformatics.org/sms2/orf_find.html). We then used tblastn (https://blast.ncbi.nlm.nih.gov/Blast.cgi) to map putative Y-linked peptides to the female genome assembly for each species, to identify whether Y-linked transcripts had paralogous sequences in the female genome. We ignored all transcripts that aligned poorly with a BLAST score below 50 or with less than 40% sequence identity. We then used the software exonerate[62] with parameters: —exhaustive, protein2-genome, –n 1, to extract coding sequences for the Y-linked transcripts as well as their paralogs in the female genomes and then aligned Y-linked transcripts to the coding sequences of these paralogs using the software prank (http://wasabiapp.org/software/prank). Finally, we used KaKs_Calculator[63] to determine Ka, Ks and Ka/Ks values. Sequences of paralogs of putative Y-linked transcripts are given in Supplementary Data 2.

For the four species in our analysis with neo-sex chromosomes, we used homology to *D. melanogaster* as well as their published genome assemblies to determine the chromosomal location of paralogs of putative Y-linked transcripts. To this end, we mapped female genomic scaffolds to coding sequences from *D. melanogaster* and the published species genomes using BLAT with default parameters and then used the best alignment to assign paralogs to chromosomal arms. We classified transcripts as being autosomal, neo-X-linked, X-linked or ancestrally Y-linked (i.e., genes that are Y-linked in *D. melanogaster* or in *D. pseudoobscura*, for *D. miranda*) based on the chromosomal location that we determined for their paralogs. Paralogs whose chromosomal location could not be identified were placed in the 'unknown' category.

**Coverage analysis for paralogs of putative Y-linked transcripts**. We used previously published genomic coverage data as well as genome assemblies[9] to determine the coverage of the genomic scaffolds that the paralogs of the putative Y-linked transcripts in the female genome are located on. We then plotted a histogram of log2(Normalized Male/Female) coverage for all genomic scaffolds highlighting the coverage of the scaffolds containing the Y-linked paralog in red lines (Fig. 4).

For *D. busckii*, no published coverage data were available. We used SOAP *de novo* to build a genome assembly from male and female genomic reads and then aligned male and female genomic reads separately to the *de novo* assembled genome using bowtie2[57] with default parameters. We then used soapcoverage to calculate male and female genomic coverages for all scaffolds whose length was at least 1000 bp. We then proceeded similarly to the other species in the analysis to obtain a coverage histogram.

**Tissue-specific expression**. For the six species in our analysis for which we had RNA-seq data from male and female heads, ovaries, and testes (Supplementary Table 2), we calculated expression of the Y-linked transcripts for each tissue as

TPM (transcripts per million) values using the software kallisto[64] with default parameters.

**Tissue-specific expression of Y homologs in *D. melanogaster***. For each species, we used BLAT[56] with a translated nucleotide and a translated database to identify the *D. melanogaster* genes that are homologous to the putative Y-linked transcripts using default parameters and a BLAT score cutoff of 50. We then downloaded tissue-specific expression profiles for each of those genes from flybase.org, in order to investigate the spatio-temporal expression patterns of genes in *D. melanogaster* whose homologs have become Y-linked in the different fly species investigated.

**Data availability**. Newly collected data have all been uploaded to GenBank. Supplementary Table 2 gives an overview of all the data sets used, including accession numbers for newly collected sequences (Bioproject PRJNA385725 [SRX2822436- SRX2822453] and SRX2788162).

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

# ARTICLE

25. Hall, A. B. et al. Six novel Y chromosome genes in *Anopheles* mosquitoes discovered by independently sequencing males and females. *BMC. Genomics* **14**, 273 (2013).

26. Carvalho, A. B. & Clark, A. G. Efficient identification of Y chromosome sequences in the human and *Drosophila* genomes. *Genome Res.* **23**, 1894–1907 (2013).

27. Hall, A. B. et al. Radical remodeling of the Y chromosome in a recent radiation of malaria mosquitoes. *Proc. Natl Acad. Sci. USA* **113**, E2114–E2123 (2016).

28. Krsticevic, F. J., Santos, H. L., Januário, S., Schrago, C. G. & Carvalho, A. B. Functional copies of the *Mst77F* gene on the Y chromosome of *Drosophila melanogaster*. *Genetics.* **184**, 295–307 (2010).

29. Daines, B. et al. The *Drosophila melanogaster* transcriptome by paired-end RNA sequencing. *Genome Res.* **21**, 315–324 (2011).

30. Carvalho, A. B. & Clark, A. G. Y chromosome of *D. pseudoobscura* is not homologous to the ancestral *Drosophila* Y. *Science* **307**, 108–110 (2005).

31. Kong, L. et al. CPC: assess the protein-coding potential of transcripts using sequence features and support vector machine. *Nucleic Acids Res.* **35**, W345–W349 (2007).

32. Krzywinska, E., Dennison, N. J., Lycett, G. J. & Krzywinski, J. A maleness gene in the malaria mosquito *Anopheles gambiae*. *Science* **353**, 67–69 (2016).

33. Holt, R. A. et al. The genome sequence of the malaria mosquito *Anopheles gambiae*. *Science* **298**, 129–149 (2002).

34. Larracuente, A. M., Noor, M. A. F. & Clark, A. G. Translocation of Y-linked genes to the dot chromosome in *Drosophila pseudoobscura*. *Mol. Biol. Evol.* **27**, 1612–1620 (2010).

35. Sturgill, D., Zhang, Y., Parisi, M. & Oliver, B. Demasculinization of X chromosomes in the *Drosophila* genus. *Nature.* **450**, 238–241 (2007).

36. Assis, R., Zhou, Q. & Bachtrog, D. Sex-biased transcriptome evolution in *Drosophila*. *Genome Biol. Evol.* **4**, 1189–1200 (2012).

37. Lindsley, D. L., Roote, J. & Kennison, J. A. Anent the genomics of spermatogenesis in *Drosophila melanogaster*. *PLoS ONE* **8**, e55915 (2013).

38. Texada, M. J., Simonette, R. A., Johnson, C. B., Deery, W. J. & Beckingham, K. M. *yuri gagarin* is required for actin, tubulin and basal body functions in *Drosophila* spermatogenesis. *J. Cell. Sci.* **121**, 1926–1936 (2008).

39. Lahn, B. T., Pearson, N. M. & Jegalian, K. The human Y chromosome, in the light of evolution. *Nat. Rev. Genet.* **2**, 207–216 (2001).

40. Kaiser, V. B., Zhou, Q. & Bachtrog, D. Nonrandom gene loss from the *Drosophila miranda* neo-Y chromosome. *Genome Biol. Evol.* **3**, 1329–1337 (2011).

41. Landeen, E. L., Muirhead, C. A., Wright, L., Meiklejohn, C. D. & Presgraves, D. C. Sex chromosome-wide transcriptional suppression and compensatory cis-regulatory evolution mediate gene expression in the *Drosophila* male germline. *PLoS Biol.* **14**, e1002499 (2016).

42. Vibranovski, M. D. Meiotic sex chromosome inactivation in *Drosophila*. *J Genomics* **2**, 104–117 (2014).

43. Ritossa, F. in *Genetics and Biology of Drosophila* (eds Ashburner, M. & Novitski, E.) 801–846 (Academic Press, 1976).

44. Marchi, A. & Pili, E. Ribosomal RNA genes in mosquitoes: localization by fluorescence *in situ* hybridization (FISH). *Heredity (Edinb)* **72**(Pt 6): 599–605 (1994).

45. Brianti, M. T., Ananina, G., Recco-Pimentel, S. M. & Klaczko, L. B. Comparative analysis of the chromosomal positions of rDNA genes in species of the *tripunctata* radiation of *Drosophila*. *Cytogenet. Genome. Res.* **125**, 149–157 (2009).

46. Bedo, D. G. & Webb, G. C. Conservation of nucleolar structure in polytene tissues of *Ceratitis capitata* (Diptera: Tephritidae). *Chromosoma.* **98**, 443–449 (1989).

47. Stuart, W. D., Bishop, J. G., Carson, H. L. & Frank, M. B. Location of the 18/28S ribosomal RNA genes in two Hawaiian *Drosophila* species by monoclonal immunological identification of RNA.DNA hybrids *in situ*. *Proc. Natl Acad. Sci. USA* **78**, 3751–3754 (1981).

48. Willhoeft, U. Fluorescence *in situ* hybridization of ribosomal DNA to mitotic chromosomes of tsetse flies (Diptera: Glossinidae: Glossina). *Chromosome Res.* **5**, 262–267 (1997).

49. Roy, V. et al. Evolution of the chromosomal location of rDNA genes in two *Drosophila* species subgroups: *ananassae* and *melanogaster*. *Heredity (Edinb)* **94**, 388–395 (2005).

50. Coen, E. S. & Dover, G. A. Unequal exchanges and the coevolution of X and Y rDNA arrays in *Drosophila melanogaster*. *Cell* **33**, 849–855 (1983).

51. Lin, F., Xing, K., Zhang, J. & He, X. Expression reduction in mammalian X chromosome evolution refutes Ohno's hypothesis of dosage compensation. *Proc. Natl Acad. Sci. USA* **109**, 11752–11757 (2012).

52. Julien, P. et al. Mechanisms and evolutionary patterns of mammalian and avian dosage compensation. *PLoS Biol.* **10**, e1001328 (2012).

53. Luo, R. et al. SOAPdenovo2: an empirically improved memory-efficient short-read *de novo* assembler. *Gigascience* **1**, 18 (2012).

54. Grabherr, M. G. et al. Full-length transcriptome assembly from RNA-Seq data without a reference genome. *Nat. Biotechnol.* **29**, 644–652 (2011).

55. Kim, D. et al. TopHat2: accurate alignment of transcriptomes in the presence of insertions, deletions and gene fusions. *Genome. Biol.* **14**, R36 (2013).

56. Kent, W. J. BLAT--the BLAST-like alignment tool. *Genome Res.* **12**, 656–664 (2002).

57. Langmead, B. & Salzberg, S. L. Fast gapped-read alignment with Bowtie 2. *Nat. Methods.* **9**, 357–359 (2012).

58. Pertea, G. et al. TIGR Gene Indices clustering tools (TGICL): a software system for fast clustering of large EST datasets. *Bioinformatics.* **19**, 651–652 (2003).

59. Surget-Groba, Y. & Montoya-Burgos, J. I. Optimization of *de novo* transcriptome assembly from next-generation sequencing data. *Genome Res.* **20**, 1432–1440 (2010).

60. Roberts, A. & Pachter, L. Streaming fragment assignment for real-time analysis of sequencing experiments. *Nat. Methods* **10**, 71–73 (2013).

61. Koch, P., Platzer, M. & Downie, B. R. RepARK--*de novo* creation of repeat libraries from whole-genome NGS reads. *Nucleic Acids Res.* **42**, e80 (2014).

62. Slater, G. S. C. & Birney, E. Automated generation of heuristics for biological sequence comparison. *BMC. Bioinformatics* **6**, 31 (2005).

63. Zhang, Z. et al. KaKs_Calculator: calculating Ka and Ks through model selection and model averaging. *Genomics Proteomics Bioinformatics* **4**, 259–263 (2006).

64. Bray, N. L., Pimentel, H., Melsted, P. & Pachter, L. Near-optimal probabilistic RNA-seq quantification. *Nat. Biotechnol.* **34**, 525–527 (2016).

## Acknowledgements

Funded by NIH grants (R01GM076007, GM101255 and R01GM093182) to D.B. We thank Beatriz Vicoso for helpful discussions on this project and Zaak Walton, Matthew Nalley and Alison Nguyen for technical assistance.

## Author contributions

S.M. and D.B. conceived the study, collected and analyzed the data and wrote the manuscript.

## Additional information

**Competing interests:** The authors declare no competing financial interests.

