## [Peer Review File · Nature Communications]

Reviewers' comments:

Reviewer #1 (Remarks to the Author):

In this paper the authors study evolution of Y chromosome gene content across Diptera. To do this they first construct a subtraction pipeline to identify Y linked genes, by comparing female genome data to transcriptome data from males and females. A similar protocol has previously successfully been applied to identify Y linked genes in mammals. The authors validate the method by testing on *Drosophila melanogaster*, where it identifies all currently known Y linked genes, apart from the newly acquired gene FDY that their pipeline omits because of the gene's high sequence similarity to its autosomal paralog. The method also identifies a novel *D. melanogaster* Y linked gene. The method is then applied to a suite of other Diptera species. The ancestry and function of identified putatively Y linked genes are used to draw conclusions with respect to general patterns of Y chromosome gene evolution in this taxa. A strength with the study is that the authors can compare evolution at independently derived Y chromosomes, and at Y chromosomes of different ages. The main findings are that i) gene content is rapidly reduced on newly formed Y chromosomes, ii) genes with male functions are retained longer than other genes on newly formed Y chromosomes, and iii) eventually 'all' original genes are lost, and current genes on old Y chromosomes have 'all' been derived from elsewhere in the genome. These results are largely in line with theory and what previously has been discussed for *Drosophila*, and emphasizes differences previously found between flies and mammals. The fact that this study is conducted on such a broad taxonomic scale within Diptera does however make these results novel. I thus think this study adds substantially to our empirical knowledge with respect to evolution of Y chromosomes.

Comments:

I believe the data the authors present for the specificity and sensitivity of the subtraction pipeline used to identify Y linked genes in *D. melanogaster* is convincing. When applied to other species the results look more variable, and the cut-off of only including species with twice or more identified male than female specific genes seems very arbitrary. I wonder if the authors could expand a bit more on this and discuss why so many "female specific" genes are discovered in some species.

In the manuscript there is a focus on that genes which have been transposed/translocated to the Y chromosome have an autosomal ancestry. I am, however, not sure if any evidence is provided for that the X chromosome is not contributing to the extent expected from its relative size. It is of course difficult to distinguish between genes that have been retained on the Y from when it was an autosome from those genes that have been copied and moved to the Y from the X secondarily, but this is nothing which is discussed.

It has been suggested that the origin of the *D. melanogaster* Y chromosome could be a B chromosome, which have accumulated genes through transposition/translocation. Can this study put an end to this speculation?

Why are *Eutreta diana* and *Tephritis californica* listed in the tables as studied species? Since

they presumably have ZW sex chromosomes it does make sense to include them in a paper focusing on Y chromosome evolution. If the study focused on both Y and W chromosomes it would make sense, but then *E. diana* should be included according to the authors cut off of including species having twice as many, in this case, female to male contigs.

Did the three Y linked genes identified for *Anopheles gambiae* overlap with the 8 genes identified in the more comprehensive study by Hall et al 2016 on the *Anopheles gambiae* Y chromosome?

Clogmia albipunctata appears to have a paralog. Why is this species not included in fig 3?

The authors find many more putatively Y-linked non-coding than coding transcripts. Why is this? Is this an interesting finding or a potential artifact? Please expand.

Last line pg 2. Remove “;”

Pg. 8. There is no fig 4A as I understand, only fig 4.

Pg. 9. There is no fig 4B as I understand, only fig 4.

Figures & Tables

Figures and legends need improvement.

Figure 1. A) Please add scale and specify what the grey boxes are. B) Please add more detail to legend. Also, why are the stocks tested mostly different for males and females? C) Specify that colors refers to expression level and explain why the scale is so different between bar 1 and 2. Abbreviations need to be explained.

Figure 2. Why different grey/sand colors for the different species?

Figure 3. I presume the distributions represent male to female genomic read counts. Given that the left peaks in the bimodal distributions are centered around -1 (X linked genes?) I suppose the X axis is $\log_2(\text{male/female})$. If this is correct please spell this out in the legend and name the X axis. To make it easier for the reader it would also help to say that autosomal paralogs are expected to have a value of approx. 0 and X linked a value of approx. -1. For species with few Y linked paralogs it is easy to see the male/female expression ratio, but for genes with many it gets very messy. As of now it looks like the *D. miranda* paralogs are more or less evenly distributed, but I guess this is not the case. Would it be possible to display these as histograms overlaying the transcript histograms, to get a better feel for their distribution?

Figure 4. Marks indicating K_a , K_s and K_a/K_s values have very low resolution.

Figure 5. Explain what the Z- scores stand for here.

Figure S2. Please specify what RED and BLUE depicts and what the multiple blue lines stand

for.

Figure S3. This figure does not seem to represent what is implied from the sentence at end of page 4. It seems like they authors mean fig S4. But, then there is no reference in text to fig. S3.

Reviewer #2 (Remarks to the Author):

Y-chromosomes undergo peculiar evolutionary transitions, including gene inactivation, gene loss and the accumulation of repetitive elements. These processes create substantial challenges to current sequencing approaches, and consequently our knowledge of the gene content of Y-chromosomes is severely limited. To date, only a few Y-chromosomes' gene content has been studied in detail. Here, the authors combine both genomic and transcriptomic data of males and females from several (dipteran) fly species to identify sequences that are male-limited and hence may be located on the Y-chromosome. They verify the efficiency of this approach with available information from *Drosophila melanogaster*, in which they are able to identify all currently-known Y-linked genes as well as a novel gene. From this, they continue by identifying candidate Y-linked genes in other Dipteran insects. In this insect group, different chromosomes have become Y-chromosomes at different times in the past. The authors compare gene content of such Y-chromosomes to their homologous counterparts in *D. melanogaster*. Through this, they show that Y-chromosome gene content is primarily shaped by translocation from autosomes onto the Y-chromosome, rather than reflecting ancient gene content of the autosomal progenitor chromosome, such as seen in mammals. The approach used is very elegant (in fact, it is a prime example of the utility of comparative genomics in evolutionary studies), and may prove useful in future studies of Y-chromosomal gene content. Nonetheless, the manuscript comes short in other aspects, such as taking into account the evolutionary relevance of the data that are used here. Thus, although it is an exciting piece of scientific work, the paper requires major revisions before it can be accepted for publication. It might be wise to consider submitting it to a different journal whose focus is more towards bioinformatics/genomics as the manuscript currently leans on the technical relevance of the work.

Major comments:

Throughout the manuscript, the authors compare between a given species and *D. melanogaster*. However, *D. melanogaster* is by no means a basal species in the dipteran phylogeny, and hence the comparison is not adequate from an evolutionary perspective. Given its role as a model organism, this approach is not strange, but by itself the conclusions that are drawn by this comparison might be invalidated by other species. In what way do the data from other species with regard to e.g. gene location validate the results as acquired by the comparison between *D. melanogaster* and another species? That is, Y-linked genes are considered to have been present on the Y before it became a sex chromosome if it has a homologous gene in *D. melanogaster* on the same Muller element

(i.e. the chromosome that is Y in species A is an autosome in species B). It is considered to have originated from a different chromosome if the gene is located on a different Muller element. A caveat in this approach is that genes need not remain on the same autosome between species. What levels of autosome-autosome movement can be distinguished between species? That is, how many genes are preserved on the same chromosome in species A and B, and how many are located on different Muller elements in species A and B?

More information regarding this issue may be obtained by screening for chromosomal locations in species other than *D. melanogaster*, and determine whether gene locations in *D. melanogaster* are representative of the basal evolutionary state of a gene or not. This would be required to validate the conclusions regarding genes that are 'ancestrally' located on a given Muller element and those that have been recruited onto this element from a different location.

Are the locations of genes with regard to Muller elements supported by other species? That is, are genes that are inferred to have been located on the Y-chromosome when it was still an autosome always on the same Muller element in different species, or do they map to different Muller elements? If a gene is predominantly mapped to a different Muller element, then this would argue against a common precursor Y-chromosome. Can it then be excluded that convergent evolution in *D. melanogaster* and the focal species occurred, i.e. a gene translocated to the same Muller element in the two species, but these Muller elements evolved differently over time? Conclusions would be considerably solidified if they are supported by data derived from comparisons to other (more basal) species.

Gene expression of Y-linked genes is commonly found to be high in male-specific tissues (e.g. testes), which the authors state is due to male-specific selective pressures. The authors do not assay these selective forces, but rather infer them from these observations. This creates a bit of a risk in that genes may be male-specifically selected because they are on the Y-chromosome (and hence are male-limited) or they are on the Y-chromosome because they are male-specifically selected. Some more clarification could be applied at various points in the manuscript regarding these alternative scenarios.

Many of the figures and tables are difficult to understand, and even more so for anyone who is not acquainted with comparative genomics. Virtually none of them are easily understood as stand-alone entities, and more information should be added in the figure and table captions. Further comments regarding specific figures:

- Fig. 1A: What is the total size of this locus? Please include.
- Fig. 1C: What are the different stages in C? What is on the legend? How are these scores derived?
- Fig. 2: How are these Muller elements defined? Please include a karyogram (e.g. from *D. melanogaster*) for non-expert readers. The bottom branches (*D. busckii* and *D. albomicans*) are poorly aligned; also, does *D. albomicans* have a doubly-fused neo-sex chromosome pair? Why are all *Bactrocera* 'yellow', except for *E. hians*? What do the grey horizontal lines signify? Why are drawn images of some species included (and others not)?
- Fig. 3: What is on the X-axis in each figure? How should these be interpreted? For

example, what does a double peak such as seen in e.g. *D. miranda* signify? Please do not extend data beyond the axis (e.g. *C. trivittatus*).

- Fig. 4: The text mentions a figure 4B (on p. 9), but this is not included; likewise there is a reference to 4A on p. 8.
- Fig. 5: Reorder to match phylogeny (from Fig. 2)
- Fig. S2: What is on the X-axis? Transcript size in bp?
- Fig. S3: Define 'high' and 'low' expression (relative to cell cycle gene expression?)
- Fig. S4: Is not referred to in the text. Also, what is the relevance of this figure?
- Table S1: Remove bottom entry?
- Table S2 to S5: Please sort by species name or other logical variable (they appear to be randomly ordered).
- Please pay attention to species names in the tables. Some of them are misspelled. For example Table S3: '*Sarcophaga bulata*' should be '*Sarcophaga bullata*'; '*Bactrocera olea*' should be '*Bactrocera oleae*'.

Minor comments:

P2: second-to-last line: "provided" should be "proven".

2nd paragraph: 'Diptera flies' should be 'Diptera' or 'Dipteran flies', as Diptera solely comprises flies.

Missing closing bracket at end of 2nd paragraph.

P5: What is known about kirre expression in males? Is it female-specifically expressed, or is it also expressed in males but at a too low level to be detected? This might be relevant to figure out why it is picked up in your pipeline.

"Several studies have shown that X chromosomes in *Drosophila* are an unpreferred location for genes with male-function and all three genes that have been lost are expressed predominantly in testis (both in *D. melanogaster* and *D. pseudoobscura*)." Please provide pr

Reviewer #3 (Remarks to the Author):

The evolution of sex chromosomes is a fascinating topic in general and there are well-developed theories as to how and why a Y-chromosome loses genes as the neoX and neoY diverge from an ancestral autosome pair. There are also cases of gene movement on and off both the X and Y. In the case of the Y, immediate selection in males should promote the movement of genes with male functions to the Y, despite the dangers of mutation accumulation due to lack of recombination. In mammals, this is accommodated by duplication of Y-linked genes for example. Using genomics data, especially in the insects, has helped confirm/refute many of the longstanding theories on sex chromosome evolution. In this paper, the authors have looked at a range of insect species and have come to some strong conclusions. I think that more work is needed to make the case they present. My

major concern is that sex-specific expression and reference to the genome assembly is simply not good enough evidence of Y-linkage. There are many genes that are expressed in just one sex and there are likely to be many holes in the assemblies of the species used in the manuscript. As a result many of the called Y-linked genes could be from somewhere else.

1. What are the assembly statistics on the various organisms? This information is critical for evaluating the methodology. In the case of *Teleopsis dalmanni*, NCBI reports 174,639 scaffolds! Many genes expressed in males but not mapping to a poor assembly are false positives. Using the well-assembled *Drosophila melanogaster* genome as a test is not appropriate.

2. The candidate Y genes need to be validated. Southern blotting using DNA from females versus males or FISH on chromosomes could be used to determine which of the candidate Y-linked genes are actually on that chromosome. These or some alternative secondary method for validating linkage is needed.

Reviewer #4 (Remarks to the Author):

ms title: Convergent evolution of Y chromosome gene content in flies

Authors: Shivani Mahajan & Doris Bachtrog

journal: Nature Communications

The main findings of the paper, according to the authors, can be summarized as follows:

1- They developed a computational pipeline that has high sensitivity and specificity to identify Diptera Y genes (Fig S1).

2. Diptera Y-linked genes were acquired from the autosomes, instead of being relics from the original autosome pair that originated the sex-chromosomes (as happens in mammals). This is Fig 2 of the paper.

3. This acquisition occurred gradually (Fig 4).

4. These genes already have male-related functions before being acquired by the autosomes (Fig 5, part).

5. In neo-Y species, male genes are preferentially retained (Fig. 5 part).

Diptera Y chromosomes have been relatively well studied in *Drosophila*, and, to a lesser extent, in mosquitoes (*Anopheles*, *Aedes*). Many species studied in this paper belong to Diptera families that have not been investigated before, and this taxonomic breadth is, in my opinion, the main strength and interest of the paper. The authors found a novel gene in the *D. melanogaster* Y, a good candidate for the sex-determining gene in *Chaoborus trivittatus*, and 184 potentially Y-linked protein-coding genes in all species (although more than a hundred of them came from neo-Y chromosomes).

I saw two major problems in the paper. First, perhaps as a consequence of the rather large

number of species studied, it is a bit "too fast" and less solid than would be desirable, and hence some conclusions may not hold under a more careful analysis, if this is done in the future. For example, as detailed below, the computational pipeline produce many nonsensical results and false-positives (genes that are present in the XX female genome and absent in the XY male genome), and hence it is almost unavoidable that a sizeable portion of the reported "184 protein--coding transcripts" turn out to be artifacts. Most people in the field would feel uncomfortable to assign a gene to the Y chromosome without a PCR confirmation of Y-linkage (true Y-genes will amplify only in male DNA), and some additional investigation of potential function (e.g., to exclude pseudogenes), etc. Although the authors initially refer to these 184 genes as "potentially Y-linked", the cautionary "potentially" quickly disappear from the text, and the authors draw bold conclusions about Y-chromosome evolution from a dataset and analysis what may be more precisely described as a preliminary and exploratory. The authors repeatedly alluded to the *D. melanogaster* results (they found nearly all previously found genes, plus a novel one) as a confirmation that their computational pipeline has "high sensitivity and specificity" but this is directly contradicted by a look at Table S5. Another example of quick and less solid approach was the use of divergence with uncurated paralogs to support the conclusion that Y-linked genes were acquired gradually. Knowledge about Y-chromosomes lag behind the other chromosomes for good reasons: they present many difficulties to study, which might recommend a more careful work. On the other hand, it is difficult to be comprehensive in two dimensions (i.e., number of species studied and depth of analysis in each one), and it may be the case that the weaknesses in one dimension are justified by the strength in the other. This is not my opinion, but the adequate stringency level also depends on the scope of the journal, so I think that this is a judgment that the Editor will be better positioned to do.

One way to look at the above problems of reliability is that the conclusions of the paper probably are correct because most of them confirm in a larger taxonomic sample what has been found before in more detailed work done before with *Drosophila* and, to a lesser extent, in mosquitoes. However, it is possible that several Diptera families turn out to "not follow" the *Drosophila* case, and a more careful study may disclose new phenomena (actually, this is the main justification of studying a broad taxonomic sample). One bad consequence of quick studies is that they in practice discourage subsequent careful work, even by the same group. This brings the second major problem I mentioned: although the findings of the authors suggest that Y chromosomes from distant Diptera families evolve very similarly to the pattern well documented in *Drosophila*, this similarity is basically not acknowledged in the paper, as even a cursory read of the Conclusions demonstrate. For these omissions I see no justification within the scope of good science.

Thus, in my opinion there are important methodological problems in the paper which would prevent its acceptance in its present form. I also think that the presentation and discussion of the results is misleading at several passages, either by presenting as new well known results obtained other groups, or by failing to openly acknowledge the limitations of the methods and analysis employed. I detailed below these problems, in the same order of the five main findings presented above. An unrelated problem is that I could not find in the Nat. Comm. website the "Supplementary file 1", which is essential for fully reviewing the

paper (it contains the sequence of the genes found in the Y chromosome of all species). I ask the Editor to double-check, may be I am missing something.

1. Computational pipeline.

Methodological issue: The pipeline has many arbitrary parameters. How were they chosen? In particular, was *D. melanogaster* used to fine-tune the pipeline parameters? If yes, then it is not appropriate to use this species to evaluate the pipeline efficiency, because it amounts to a sort of over-fitting. This information should be included in the Methods section.

Data versus conclusions: In several paragraphs the authors stated that "Overall, our pipeline shows both high sensitivity and specificity for detecting Y-linked genes, especially for species and genomic regions with high read coverage". However, in 9 out of 24 species it did not yield any candidate, or yield too many non-sensical results (as defined by the authors, "too many" means more than 1/3 of the transcripts matching the XX female genome but not XY males; Table S5). Even among the 15 species that were analysed, there is a fair number of "female-specific" transcripts which are false positives, since female DNA is a subset of the male DNA. For example, in *E. hians* (which has a good transcriptome dataset), the authors reported 28 male specific and 9 female specific transcripts. Hence, we would expect that among these 28 transcripts (deemed as Y-linked), 9 would be false-positives. Failure in 9 out of 24 species, and an estimated rate of false positives of 9 out of 28 transcripts are not what most people will understand by high sensitivity and specificity; it is more likely an exploratory tool. There is no major problem on using exploratory tools, unless this is not openly acknowledged.

Acknowledgement of previous work : Other groups studying Diptera Y (mosquitoes and *Drosophila*) developed specific methods to identify Y-linked genes, which were published in major genomic journals, and that do not seem to suffer from the above mentioned problems. None were discussed (or cited) in the paper.

Six novel Y chromosome genes in *Anopheles* mosquitoes discovered by independently sequencing males and females Hall et al BMC Genomics. 2013; 14: 273.

Carvalho, A.B., and A.G. Clark. Efficient identification of Y chromosome sequences in the human and *Drosophila* genomes. Genome Research, 23: 1894-1907, 2013.

For the benefit of the readers, it would be desirable that the authors discuss/compare what they propose to what is available in the literature.

2. Diptera Y-linked genes were acquired from the autosomes, instead as being relics from the original autosome pair that originated the sex-chromosomes (as happens in mammals).

Acknowledgement of previous work: the pattern found by the authors actually has been thoroughly documented in *Drosophila*, starting in 2000 (reviewed in Carvalho et al TIG 2009). For example, Carvalho Lazzaro and Clark (PNAS 2000) described their findings as follows: "A striking pattern emerges from the phylogeny of the Y dyneins: they all are closely related to other *Drosophila* genes, but none of these paralogous genes is X-linked (Fig. 4). The same pattern occurs with PRY.". Thus, the pattern initially discovered in *Drosophila* seems to be valid for Diptera in general. This similarity is at most tersely

acknowledged in the ms, or not acknowledged at all, as a cursory reading of the Conclusion section shows: Diptera and mammalian Y are compared, and external references are provided only for the mammalian work, which would induce readers to believe that this is the first paper that investigate Y-chromosome evolution in Diptera. Analogous omissions are presented in many places of the ms. What the authors really found is that Y evolution of other Diptera families seem very similar to previously documented in *Drosophila*; this should be openly acknowledged. Mosquito data is more scanty and less clear, but should also be discussed in this context.

3. The acquisition of Y-linked genes occurred gradually (Fig 4).

Acknowledgement of previous work: Similar to #2 above. This pattern was well documented in *Drosophila*. See, for example, Fig 2 of Koerich et al Nature 2008. and Fig. 4 of Carvalho and Clark Genome Research 2013.

Methodological issues: Ka Ks analysis between the Y-linked gene and a paralog was used to roughly estimate the time of duplication that generated the Y-linked genes. Hence it is essential that the choosen paralog is the correct one, which is not trivial and, I think, not possible to be done with the uncurated computational approach they used. The authors should provide a table linking each Y-linked gene to its paralog, including *D. melanogaster*. The fasta sequence of these paralogs should also be provided, since many genomes were not previously annotated.

4. This genes already have male-related functions before being aquired by the autosomes (Fig 5, part).

Acknowledgement of previous work: Similar to #2 and #3 above. This pattern was well documented in *Drosophila* since 2000:

Carvalho and Clark Genome Research 2013: " Hence, these four genes fit the general pattern of *Drosophila* Y-linked genes, formerly autosomal male-specific genes transposed to the Y-chromosome (Carvalho et al. 2009)." Again, no clue of these previous results in the ms.

other points:

Table S5: Pleasae add a column with the sex-chromosome system (XY ZW , or homomorphic). This will help the readers to understand the results.

"(though the application of long-read PacBio technology has provided useful in assembling individual Y-linked genes in *D. melanogaster*; " Actually PacBio allowed the assembly of fairly large genomic regions of the Y, containing several genes / pseudogenes .

Response to Reviewer #1:

In this paper the authors study evolution of Y chromosome gene content across Diptera. To do this they first construct a subtraction pipeline to identify Y linked genes, by comparing female genome data to transcriptome data from males and females. A similar protocol has previously successfully been applied to identify Y linked genes in mammals. The authors validate the method by testing on *Drosophila melanogaster*, where it identifies all currently known Y linked genes, apart from the newly acquired gene FDY that their pipeline omits because of the gene's high sequence similarity to its autosomal paralogs. The method also identifies a novel *D. melanogaster* Y linked gene. The method is then applied to a suite of other Diptera species. The ancestry and function of identified putatively Y linked genes are used to draw conclusions with respect to general patterns of Y chromosome gene evolution in this taxa. A strength with the study is that the authors can compare evolution at independently derived Y chromosomes, and at Y chromosomes of different ages. The main findings are that i) gene content is rapidly reduced on newly formed Y chromosomes, ii) genes with male functions are retained longer than other genes on newly formed Y chromosomes, and iii) eventually 'all' original genes are lost, and current genes on old Y chromosomes have 'all' been derived from elsewhere in the genome. These results are largely in line with theory and what previously has been discussed for *Drosophila*, and emphasizes differences previously found between flies and mammals. The fact that this study is conducted on such a broad taxonomic scale within Diptera does however make these results novel. I thus think this study adds substantially to our empirical knowledge with respect to evolution of Y chromosomes.

R1.1 - I believe the data the authors present for the specificity and sensitivity of the subtraction pipeline used to identify Y linked genes in *D. melanogaster* is convincing. When applied to other species the results look more variable, and the cut-off of only including species with twice or more identified male than female specific genes seems very arbitrary. I wonder if the authors could expand a bit more on this and discuss why so many "female specific" genes are discovered in some species.

This is a good point. We have greatly expanded our discussion on why in some species our pipeline is less successful in identifying Y-linked sequences. In particular, we fail to identify Y-linked transcripts in species that have homomorphic sex chromosomes or are XO (such as the Hessian fly), and thus simply lack male-specific sequences. We also have difficulties in reliably identifying male-specific genes in species with very large inferred genomes (i.e. >500Mb) where genome assemblies tend to be highly fragmented. However, most of our species have genome sizes considerably smaller, and our pipeline works well inferring male-specific sequences in this case. We now have also verified our pipeline in additional species with well-assembled genomes (*Anopheles* & *D. pseudoobscura*), and by male-specific PCR amplification of many putative Y-linked genes in several non-*Drosophila* fly taxa. We discuss this more clearly now on page 6 & 7.

R1.2 - In the manuscript there is a focus on that genes which have been transposed/translocated to the Y chromosome have an autosomal ancestry. I am, however, not sure if any evidence is provided for that the X chromosome is not contributing to the extent expected from its relative size. It is of course difficult to distinguish between genes that have been retained on the Y from when it was an autosome from those genes that have been copied and moved to the Y from the X secondarily, but this is nothing which is discussed.

The reviewer is correct that we are not presenting any evidence that the X chromosome does not contribute at all to the gene repertoire of the Y in fly species. If degeneration of most ancestral genes and retention of a few is the only evolutionary force shaping gene content on the Y, we would in fact expect that all the closest paralogs of Y genes are located on the X. Instead, we show that the majority of Y-linked genes are derived from autosomes. Thus, gene gain from autosomes is important in shaping gene content of old Y chromosomes in flies, yet retention (or translocation) of X genes may also play a role. We now also state that we cannot distinguish genes that have been copied and moved to the Y from the X secondarily from those that were ancestrally located on the Y based on location information alone (that is, we may overestimate the number of genes being ancestrally Y-linked). We now discuss this more clearly on page 9.

R1.3 - It has been suggested that the origin of the *D. melanogaster* Y chromosome could be a B chromosome, which have accumulated genes through transposition/translocation. Can this study put an end to this speculation?

The idea that the *Drosophila* Y chromosome is a B chromosome is based on the lack of homologous protein-coding genes on the X and Y in *D. melanogaster* (the only homologous locus is the ribosomal RNA locus). Our findings that (1) independently formed old Y chromosomes across Diptera all show very limited or no homology with the X; (2) autosomal genes have been recruited to the Y chromosomes in multiple Diptera species; (3) Y-chromosomes can rapidly lose many/most of their ancestral genes within 1-10MY; and (4) X chromosomes preferentially lose genes that are more likely to be maintained on the Y (i.e. testis-specific genes) indeed suggests that the *D. melanogaster* Y (and the Y of all the other Diptera flies investigated) is not derived from a B chromosome. Instead, it suggests that Y chromosomes simply lose genes very quickly and gain others from autosomal locations, in a diverse set of fly species. We now discuss this on page 16 (and also include a new Figure 7 to demonstrate this dynamics more clearly).

R1.4 - Why are *Eutreta diana* and *Tephritis californica* listed in the tables as studied species? Since they presumably have ZW sex chromosomes it does make sense to include them in a paper focusing on Y chromosome evolution. If the study focused on both Y and W chromosomes it would make sense, but then *E. diana* should be included according to the authors cut off of including species having twice as many, in this case, female to male contigs.

This is a good point. We now exclude our two ZW species since our focus is on Y chromosome evolution.

R1.5 - Did the three Y linked genes identified for *Anopheles gambiae* overlap with the 8 genes identified in the more comprehensive study by Hall et al 2016 on the *Anopheles gambiae* Y chromosome? *Clogmia albipunctata* appears to have a paralog. Why is this species not included in fig 3?

We now clearly state in the paper (page 7) that the three Y-linked transcripts that we obtained all map to the YG1 and YG2 genes in *Anopheles gambiae*. Other putative Y genes by Hall et al 2016 are either limited to the G3 strain studied, or have autosomal paralogs. In particular, YG3 and YG4 are exclusively Y-linked in the G3 strain according to Hall et al. 2016. YG5 was found to be located both on an autosome and on the Y using FISH by Hall et al 2016, and would not be picked up by our pipeline if the Y-linked gene is too similar to the autosomal paralog. YG6-8 were never validated by Hall et al 2016 (and it is thus unclear whether they are indeed exclusive on the Y / shared with other strains).

Exclusion of *Clogmia albipunctata* was an oversight. We have updated Figure 4 and now also include *Clogmia albipunctata*.

The authors find many more putatively Y-linked non-coding than coding transcripts. Why is this? Is this an interesting finding or a potential artifact? Please expand.

We believe that the assignment of a large number of transcripts as non-coding (or more accurately, the failure to designate them as coding) is because the software that we use to assess coding capacity is being conservative. Coding potential calculator uses a variety of biological features to assess the protein-coding potential of a transcript, including the length of open reading frames (ORF's), whether an ORF begins with a start codon and ends with an in-frame stop codon, and various features from BLASTX queries of the transcript against a non-redundant protein database. Thus, fragmented protein-coding transcripts, or short and highly

divergent proteins may be annotated as non-coding. In fact, the software called some fragmented Y linked transcripts of *D. melanogaster* non-coding even if they mapped to parts of known protein-coding Y genes. We now discuss this on page 8.

Last line pg 2. Remove “;”

Done.

Pg. 8. There is no fig 4A as I understand, only fig 4.

Changed.

Pg. 9. There is no fig 4B as I understand, only fig 4.

Changed.

Figures & Tables

Figures and legends need improvement.

Figure 1. A) Please add scale and specify what the grey boxes are. B) Please add more detail to legend. Also, why are the stocks tested mostly different for males and females? C) Specify that colors refers to expression level and explain why the scale is so different between bar 1 and 2. Abbreviations need to be explained.

We have updated this Figure and the legend according to the reviewer's suggestion. We added a scale, and specify the grey boxes. Strains were chosen at random from the NCBI SRA database, so they differ between males and females (this is now mentioned in the Table). In Fig 2C, the values plotted are taken from flybase and represent actual RPKM values; the difference in scale is presumably due to the testis-specific expression of this gene in flies (and thus lower RPKM values in whole flies / pupae). The color key represents RPKM with black meaning no expression and green showing the highest RPKM value for expression; this is indicated in the legend now.

Figure 2. Why different grey/sand colors for the different species?

We have modified that Figure. The different shadings denote different families, and only white and grey shading are used now.

Figure 3. I presume the distributions represent male to female genomic read counts. Given that the left peaks in the bimodal distributions are centered around -1 (X linked genes?) I suppose the X axis is $\log_2(\text{male/female})$. If this is correct please spell this out in the legend and name the X axis. To make it easier for the reader it would also help to say that autosomal paralogs are expected to have a value of approx. 0 and X linked a value of approx. -1. For species with few Y linked paralogs it is easy to see the male/female expression ratio, but for genes with many it gets very messy. As of now it looks like the *D. miranda* paralogs are more or less evenly distributed, but I guess this is not the case. Would it be possible to display these as histograms overlaying the transcript histograms, to get a better feel for their distribution?

We have completely modified the Figure and the legend, according to the reviewer's suggestion. We clearly spell out in the legend that scaffolds that are X-linked have reduced male/female coverage ratio ($\log_2(\text{Mcov/Fcov})$ around -1), and are shown in yellow shading in the Figure; autosomal scaffolds (shown in grey shading) have similar coverage in males and females ($\log_2(\text{Mcov/Fcov})$ around 0). We also display the genomic coverage of paralogs in *D. miranda* as a histogram now in this Figure.

Figure 4. Marks indicating K_a , K_s and K_a/K_s values have very low resolution.

We have redone Figure 5 at higher resolution.

Figure 5. Explain what the Z- scores stand for here.

Expression values were calculated as TPM (transcript per million) using *kallisto* and were row normalized to obtain the z-scores to have a mean of 0 and standard deviation of 1 using the built-in `scale='row'` argument in the `heatmap.2` function from the package `gplots` in R. This is now spelled out in the legend.

Figure S2. Please specify what RED and BLUE depicts and what the multiple blue lines stand for.

We now explain that RED lines depict the annotated longest transcript for each gene in the legend, and the blue line depict our de novo annotated transcripts using our bioinformatics pipeline. Multiple blue lines are different isoforms for the transcripts that were assembled using our pipeline.

Figure S3. This figure does not seem to represent what is implied from the sentence at end of page 4. It seems like they authors mean fig S4. But, then there is no reference in text to fig. S3.

We fixed this error.

Response to Reviewer #2:

Y-chromosomes undergo peculiar evolutionary transitions, including gene inactivation, gene loss and the accumulation of repetitive elements. These processes create substantial challenges to current sequencing approaches, and consequently our knowledge of the gene content of Y-chromosomes is severely limited. To date, only a few Y-chromosomes' gene content has been studied in detail. Here, the authors combine both genomic and transcriptomic data of males and females from several (dipteran) fly species to identify sequences that are male-limited and hence may be located on the Y-chromosome. They verify the efficiency of this approach with available information from *Drosophila melanogaster*, in which they are able to identify all currently-known Y-linked genes as well as a novel gene. From this, they continue by identifying candidate Y-linked genes in other Dipteran insects. In this insect group, different chromosomes have become Y-chromosomes at different times in the past. The authors compare gene content of such Y-chromosomes to their homologous counterparts in *D. melanogaster*. Through this, they show that Y-chromosome gene content is primarily shaped by translocation from autosomes onto the Y-chromosome, rather than reflecting ancient gene content of the autosomal progenitor chromosome, such as seen in mammals. The approach used is very elegant (in fact, it is a prime example of the utility of comparative genomics in evolutionary studies), and may prove useful in future studies of Y-chromosomal gene content. Nonetheless, the manuscript comes short in other aspects, such as taking into account the evolutionary relevance of the data that are used here. Thus, although it is an exciting piece of scientific work, the paper requires major revisions before it can be accepted for publication. It might be wise to consider submitting it to a different journal whose focus is more towards bioinformatics/genomics as the manuscript currently leans on the technical relevance of the work.

Throughout the manuscript, the authors compare between a given species and *D. melanogaster*. However, *D. melanogaster* is by no means a basal species in the dipteran phylogeny, and hence the comparison is not adequate from an evolutionary perspective. Given its role as a model organism, this approach is not strange, but by itself the conclusions that are drawn by this comparison might be invalidated by other species. In what way do the data from other species with regard to e.g. gene location validate the results as acquired by the comparison between *D. melanogaster* and another species? That is, Y-linked genes are considered to have been present on the Y before it became a sex chromosome if it has a homologous gene in *D. melanogaster* on the same Muller element (i.e. the chromosome that is Y in species A is an autosome in species B). It is considered to have originated from a different chromosome if the gene is located on a different Muller element. A caveat in this approach is that genes need not remain on the same autosome between species. What levels of autosome-autosome movement can be distinguished between species? That is, how many genes are preserved on the same chromosome in species A and B, and how many are located on different Muller elements in species A and B?

The reviewer is correct that for some of our analysis of gene location (shown in Figure 3) we rely on conserved gene synteny between species (however, Figure 4 does not rely on that assumption, see below). Also, autosome-autosome movements of genes have no influence on any of the conclusions in our paper; the only movements that are relevant for our paper are movements between autosomes and the X chromosomes. In particular, part of our analysis focuses on whether Y-linked genes were derived from genes that were present on the chromosome pair that became the X and the Y chromosome in a focal species, or whether they are derived from an autosome (any autosome). While Figure 3 uses synteny information for this inference, Figure 3 estimates X-linkage for paralogs of Y genes independently using genomic coverage information (see below).

On one hand, there is a lot of previous work showing that gene synteny is conserved between chromosomal arms in flies. Within the *Drosophila* genus, gene movements between Muller elements are rare (Muller HJ (1940) *The New Systematics*), and chromosome synteny, to a large extent, is even conserved between the very distant groups Nematocera and Brachycera (as revealed by comparative genome sequence analysis between *D. melanogaster* and *Anopheles gambiae* [Holt et al. *Science* 2002] and dozens of more recently released fly genomes such as tse flies, house flies, *Culex*, Hessian flies etc.; see flybase.org).

Additionally, in our previous manuscript (where we showed that there are numerous transitions of sex chromosomes in Diptera; Vicoso & Bachtrög, *PLoS Biology* 2015), we did a lot of work to show that gene synteny is indeed conserved among the species investigated here, and that for the vast majority of genes we were correct in inferring their location on sex chromosomes vs. autosomes based on gene synteny inference in *D. melanogaster*. That is, if we identified a particular Muller element to be an X chromosome in a taxon, the vast majority of genes on that chromosome were inferred to be X-linked based on our genomic coverage analysis (i.e. comparing male and female genomic coverage to distinguish between X-linked and autosomal genes), and only a small fraction of genes showed evidence of X-linkage (based on coverage) if derived from different Muller elements based on gene synteny in *D. melanogaster* (see Supporting Figure 1 and 3 from the Vicoso & Bachtrög paper for all the different species investigated). In additions, we arrived at identical conclusions if we used *Anopheles gambiae* as our reference species instead of *D. melanogaster* (see Supporting Figure 2 from Vicoso & Bachtrög, *PLoS Biology* 2015). Finally, in species with assembled genomes, our inference of sex-linkage based on coverage agreed very well with assignments based on assembled genomes (see Supporting Figure 6 from Vicoso & Bachtrög, *PLoS Biology* 2015).

Thus, all available data show that gene synteny is largely conserved across flies, validating the assumption of Muller elements to assign genes to particular linkage groups.

In addition, we also performed a completely independent analysis to test for ancestral X-linkage of putative Y genes that does not depend on conserved gene synteny between the target species and *D. melanogaster* (Figure 4). In particular, by identifying the closest paralogs of putative Y-linked genes, and determining whether they are X-linked based on male and female genomic coverage, we independently come to the same conclusion: Paralogs of many Y-linked genes have male/female coverage ratios that are typical of autosome; thus many putative Y-linked genes are derived from autosomal genes, instead of being remnants of genes ancestrally present on the sex chromosomes.

More information regarding this issue may be obtained by screening for chromosomal locations in species other than *D. melanogaster*, and determine whether gene locations in *D. melanogaster* are representative of the basal evolutionary state of a gene or not. This would be required to validate the conclusions regarding genes that are 'ancestrally' located on a given Muller element and those that have been recruited onto this element from a different location.

Are the locations of genes with regard to Muller elements supported by other species? That is, are genes that are inferred to have been located on the Y-chromosome when it was still an autosome always on the same Muller element in different species, or do they map to different Muller elements? If a gene is predominantly mapped to a different Muller element, then this would argue against a common precursor Y-chromosome. Can it then be excluded that convergent evolution in *D. melanogaster* and the focal species occurred, i.e. a gene translocated to the same Muller element in the two species, but these Muller elements evolved differently over time? Conclusions would be considerably solidified if they are supported by data derived from comparisons to other (more basal) species.

See our answer above. This was done in the Vicoso & Bachtrog paper (by comparing to *Anopheles gambiae*) and in Figure 4, which does not rely on conserved synteny among species.

Gene expression of Y-linked genes is commonly found to be high in male-specific tissues (e.g. testes), which the authors state is due to male-specific selective pressures. The authors do not assay these selective forces, but rather infer them from these observations. This creates a bit of a risk in that genes may be male-specifically selected because they are on the Y-chromosome (and hence are male-limited) or they are on the Y-chromosome because they are male-specifically selected. Some more clarification could be applied at various points in the manuscript regarding these alternative scenarios.

This is a good point. While it is commonly assumed that male-biased expression equals male-specific selection pressures, the direction is less clear. On one hand, we find that genes with testis-biased expression are more likely to survive on evolving neo-Y chromosomes (see also Kaiser et al. GBE 2011 for a more thorough analysis on that) – i.e. those genes ancestrally have male-specific expression (as inferred from expression patterns of their homologs on the neo-X or from close outgroup species) and supposedly survive on the Y because they are under male-specific selection. On the other hand (and consistent with previous findings in *D. melanogaster*; see several papers by B. Carvalho and colleagues cited in our manuscript), we also find that genes that ancestrally were testis-expressed (see Figure 6A; where we infer ancestral expression patterns of putative Y genes based on their tissue-specific expression patterns in *D. melanogaster*) are more likely to be found on Y chromosomes (and many of those genes were acquired from autosomes, see Fig. 3 & 4). Expression patterns of autosomal or X-linked paralogs of testis-expressed Y-linked transcripts in *D. pseudoobscura* further support this conclusion: Most testis-expressed genes that were acquired or maintained on the neo-Y of *D. pseudoobscura* ancestrally have testis-biased expression (as inferred based on expression patterns of their paralogs, see **Fig. S5**). Thus, maintenance or acquisition of male-specific genes appears to be an important determinant of the gene repertoire of Y chromosomes in flies. We state this more clearly on page 15.

Many of the figures and tables are difficult to understand, and even more so for anyone who is not acquainted with comparative genomics. Virtually none of them are easily understood as stand-alone entities, and more information should be added in the figure and table captions. Further comments regarding specific figures:

We modified all our figures & figure legends in order to make them easier to understand.

- Fig. 1A: What is the total size of this locus? Please include.

We included a scale bar in the Figure.

- Fig. 1C: What are the different stages in C? What is on the legend? How are these scores derived?

We explain the abbreviations for the developmental stages of (now) Fig. 2C in the legends. The scores are taken from flybase (now stated in the legend).

- Fig. 2: How are these Muller elements defined? Please include a karyogram (e.g. from *D. melanogaster*) for non-expert readers. The bottom branches (*D. busckii* and *D. albomicans*) are poorly aligned; also, does *D. albomicans* have a doubly-fused neo-sex chromosome pair? Why are all *Bactrocera* 'yellow', except for *E. hians*? What do the grey horizontal lines signify? Why are drawn images of some species included (and others not)?

We have modified this Figure (now Fig. 3). In particular, we include a karyogram from *D. melanogaster*, aligned all the species correctly, changed the coloring scheme, and note that flies with the same shading belong to the same family (indicated by the common name of that family). In *D. albomicans*, two autosomal chromosomal arms became fused and then the fused autosome fused to the sex chromosomes (this is explained in detail in the main text). Images are drawn only for a subset of species because of limited space.

- Fig. 3: What is on the X-axis in each figure? How should these be interpreted? For example, what does a double peak such as seen in e.g. *D. miranda* signify? Please do not extend data beyond the axis (e.g. *C. trivittatus*).

We have completely modified this Figure, and its legend, to make it easier to understand. We explain that double-peaks correspond to species with large heteromorphic sex chromosomes (due to hemizyosity of the X in males the Mcov/Fcov ratio is reduced for X-linked scaffolds in males), and include shading in the figure to more easily distinguish X-linked from autosomal scaffolds. This is all explained in the legend now.

- Fig. 4: The text mentions a figure 4B (on p. 9), but this is not included; likewise there is a reference to 4A on p. 8.

We fixed that.

- Fig. 5: Reorder to match phylogeny (from Fig. 2)

We ordered the figure to match the phylogeny.

- Fig. S2: What is on the X-axis? Transcript size in bp?

Yes, the x-axis is the transcript size in bp. We note that now in the legend.

- Fig. S3: Define 'high' and 'low' expression (relative to cell cycle gene expression?)

We added a legend to Fig S2 to define high and low expression (from ModEncode).

- Fig. S4: Is not referred to in the text. Also, what is the relevance of this figure?

We refer to the figure now. This Figure shows that the CG41561 gene is conserved in close relatives of *Drosophila*.

- Table S1: Remove bottom entry?

We changed Table S1 and include the SRA accession number of the *D. melanogaster* lab strain we sequenced.

- Table S2 to S5: Please sort by species name or other logical variable (they appear to be randomly ordered).

Table S2 to S5 (which are fused in Table S3 now) are ordered like the phylogeny in Vicoso &

Bachtrog (and the phylogeny with fewer taxa shown in Figure 3). This is made clear now.

- Please pay attention to species names in the tables. Some of them are misspelled. For example Table S3: 'Sarcophaga bulata' should be 'Sarcophaga bullata'; 'Bactrocera olea' should be 'Bactrocera oleae'.

Thanks for pointing out that error. We re-checked all the spelling to make sure it is correct.

Minor comments:

P2: second-to-last line: "provided" should be "proven".

Changed.

2nd paragraph: 'Diptera flies' should be 'Diptera' or 'Dipteran flies', as Diptera solely comprises flies.
Missing closing bracket at end of 2nd paragraph.

Changed.

P5: What is known about kirre expression in males? Is it female-specifically expressed, or is it also expressed in males but at a too low level to be detected? This might be relevant to figure out why it is picked up in your pipeline.

Kirre is generally a very lowly expressed transcript, and we note that kirre shows higher expression in adult female relative to male (page 6).

"Several studies have shown that X chromosomes in *Drosophila* are an unpreferred location for genes with male-function and all three genes that have been lost are expressed predominantly in testis (both in *D. melanogaster* and *D. pseudoobscura*)." Please provide pr

We added citations to this statement.

Response to Reviewer #3:

The evolution of sex chromosomes is a fascinating topic in general and there are well-developed theories as to how and why a Y-chromosome loses genes as the neoX and neoY diverge from an ancestral autosome pair. There are also cases of gene movement on and off both the X and Y. In the case of the Y, immediate selection in males should promote the movement of genes with male functions to the Y, despite the dangers of mutation accumulation due to lack of recombination. In mammals, this is accommodated by duplication of Y-linked genes for example. Using genomics data, especially in the insects, has helped confirm/refute many of the longstanding theories on sex chromosome evolution. In this paper, the authors have looked at a range of insect species and have come to some strong conclusions. I think that more work is needed to make the case they present. My major concern is that sex-specific expression and reference to the genome assembly is simply not good enough evidence of Y-linkage. There are many genes that are expressed in just one sex and there are likely to be many holes in the assemblies of the species used in the manuscript. As a result many of the called Y-linked genes could be from somewhere else.

Our method is not simply based on sex-specific expression and reference to genome assembly, but is a multi-step subtraction pipeline (modeled after recent studies in mammals) that we found works very well to identify Y-linked genes in flies (see our analysis on *D. melanogaster* data). In particular, we first align male RNA-seq reads to female assemblies (to enrich for Y-linked transcripts among the unmapped RNA-seq reads), then assemble those reads that don't map to the female assembly, and remove transcripts that either map to female RNA-seq or female DNA-seq reads. Thus, even if there are holes in the assembly, genes that are not from the Y chromosome should still be removed since they should be covered by genomic reads from females. We initially had a Figure describing this procedure as a Supplement, which we now put into the main text (Figure 1) to make this more clear.

Several pieces of evidence support the validity of our pipeline. First, we don't infer many additional genes on the well-studied *D. melanogaster* Y chromosome (and the one novel Y-linked gene that we identify was verified using genomic reads from additional strains). Secondly, when we switch the sexes and apply our pipeline to identify female-specific transcripts, we generally identify many fewer (or no) female-specific transcripts (apart from two species that lack a heteromorphic Y chromosomes, in which case we would expect a similar number of false-positives) and in two species with especially large genomes (where assemblies are highly fragmented). Thus, overall, our pipeline seems highly specific in identifying Y-linked transcripts in flies.

In addition, we now have also verified our pipeline in additional species with well-assembled genomes (*Anopheles gambiae* & *D. pseudoobscura*), and find that the vast majority of putative Y-linked genes identified by our pipeline map to either Y-linked scaffolds in these species (3 out of 3 in *Anopheles gambiae*), or unmapped scaffolds in the assembly (which are presumably Y-linked; this is the case for 11 out of 12 putative Y-linked transcripts in *D. pseudoobscura*; and 59 out of 63 putative Y-linked transcripts in *D. miranda* map either to the neo-sex chromosomes, or unplaced scaffolds).

Finally, we now also include male-specific PCR amplification of many putative Y-linked genes in several non-Drosophila fly taxa for which we had male and female samples available (a total of 24 putative Y-linked transcripts were validated by PCR in *Themira minor*; *Teleopsis dalmanni*; *Ephydra hians* and *Phortica variegata*). We discuss this more clearly now on page 7 and Table S4.

1. What are the assembly statistics on the various organisms? This information is critical for evaluating the methodology. In the case of *Teleopsis dalmanni*, NCBI reports 174,639 scaffolds! Many genes expressed in males but not mapping to a poor assembly are false positives. Using the well-assembled *Drosophila melanogaster* genome as a test is not appropriate.

Assembly statistics are given in Table S1. Note that we are not using the published assembled *D. melanogaster* genome to identify Y-linked transcripts in *D. melanogaster*, but instead, we are using exactly the same pipeline that we use for all the other species to identify Y-linked sequences. That is, we are using raw sequencing reads from *D. melanogaster* (that we collected), *de novo* genome and transcriptome assemblies (that we build from these reads), etc... - i.e. the pipeline described in Figure 1 to infer putative Y-linked genes in *D. melanogaster* and other species. Thus, the use of *D. melanogaster* is appropriate, and we make that more clear now in our paper on page 5 & 6.

2. The candidate Y genes need to be validated. Southern blotting using DNA from females versus males or FISH on chromosomes could be used to determine which of the candidate Y-linked genes are actually on that chromosome. These or some alternative secondary method for validating linkage is needed.

As stated above, we now include further verifications of our candidate Y genes. On one hand, we use mapping to well-assembled genomes in a subset of species (*Anopheles gambiae* and *D. pseudoobscura*). On the other hand, we are also using PCR to validate some of our newly inferred Y linked genes (see above). This new data and analysis is included in the manuscript on page 7 and Table S4.

Response to Reviewer #4:

The main findings of the paper, according to the authors, can be summarized as follows:

- 1- They developed a computational pipeline that has high sensitivity and specificity to identify Diptera Y genes (Fig S1).
2. Diptera Y-linked genes were acquired from the autosomes, instead as being relics from the original autosome pair that originated the sex-chromosomes (as happens in mammals). This is Fig 2 of the paper.
3. This acquisition occurred gradually (Fig 4).
4. This genes already have male-related functions before being acquired by the autosomes (Fig 5, part).
5. In neo-Y species, male genes are preferentially retained (Fig. 5 part).

Diptera Y chromosomes have been relatively well studied in *Drosophila*, and, to a lesser extent, in mosquitoes (*Anopheles*, *Aedes*). Many species studied in this paper belong to Diptera families that have not been investigated before, and this taxonomic breadth is, in my opinion, the main strength and interest of the paper. The authors found a novel gene in the *D. melanogaster* Y, a good candidate for the sex-determining gene in *Chaoborus trivittatus*, and 184 potentially Y-linked protein-coding genes in all species (although more than a hundred of them came from neo-Y chromosomes).

I saw two major problems in the paper. First, perhaps as a consequence of the rather large number of species studied, it is a bit "too fast" and less solid than would be desirable, and hence some conclusions may not hold under a more careful analysis, if this is done in the future. For example, as detailed below, the computational pipeline produce many nonsensical results and false-positives (genes that are present in the XX female genome and absent in the XY male genome), and hence it is almost unavoidable that a sizeable portion of the reported "184 protein-coding transcripts" turn out to be artifacts. Most people in the field would feel uncomfortable to assign a gene to the Y chromosome without a PCR confirmation of Y-linkage (true Y-genes will amplify only in male DNA), and some additional investigation of potential function (e.g., to exclude pseudogenes), etc. Although the authors initially refer to these 184 genes as "potentially Y-linked", the cautionary "potentially" quickly disappear from the text, and the authors draw bold conclusions about Y-chromosome evolution from a dataset and analysis what may be more precisely described as a preliminary and exploratory. The authors repeatedly alluded to the *D. melanogaster* results (they found nearly all previously found genes, plus a novel one) as a confirmation that their computational pipeline has "high sensitivity and specificity" but this is directly contradicted by a look at Table S5. Another example of quick and less solid approach was the use of divergence with uncurated paralogs to support the conclusion that Y-linked genes were acquired gradually. Knowledge about Y-chromosomes lag behind the other chromosomes for good reasons: they present many difficulties to study, which might recommend a more careful work. On the other hand, it is difficult to be comprehensive in two dimensions (i.e., number of species studied and depth of analysis in each one), and it may be the case that the weaknesses in one dimension are justified by the strength in the other. This is not my opinion, but the adequate stringency level also depends on the scope of the journal, so I think that this is a judgment that the Editor will be better positioned to do.

First, it is not necessarily the case that genomic studies that infer Y-linked genes in several taxa always refer to experimental validation. For example, the recent study of sex chromosome evolution across 17 bird taxa (Zhou et al Science 2014) only relied on next-generation sequencing to infer W-linked genes. Secondly, we now include further verifications of our candidate Y genes (see also response to Reviewer 3). On one hand, we use mapping to well-assembled genomes in a subset of species (*Anopheles gambiae* and *D. pseudoobscura*). On the other hand, we are also using PCR to validate some of our newly inferred Y linked genes. This new data and analysis is included in the manuscript on page 7 and Table S4. We also pay attention to keep "putatively" or "potentially" Y-linked throughout the text. We also include some more discussions for why it was more difficult to identify Y-linked genes in some species (mainly, these were species that either had no Y chromosome, or homomorphic sex chromosomes, or very large inferred genome sizes; see also response to Reviewer 1). We took extreme care to identify paralogous genes within the genome for our putative Y-linked genes, in order to infer ancestral locations of our candidate Y-linked genes. We agree with the reviewer that this is a tedious task, but this analysis independently confirmed our finding that the closest

paralogs of Y-linked genes are often not located on X-chromosomes, using a completely independent approach from the one relying on conserved synteny of gene content across fly species. We thus feel that our study is thorough in 'both dimensions' – careful analysis within several non-model fly species, and breadth of species, and we have taken all suggestion by the reviewers into account to further strengthen our manuscript and conclusions.

One way to look at the above problems of reliability is that the conclusions of the paper probably are correct because most of them confirm in a larger taxonomic sample what has been found before in more detailed work done before with *Drosophila* and, to a lesser extent, in mosquitoes. However, it is possible that several Diptera families turn out to "not follow" the *Drosophila* case, and a more careful study may disclose new phenomena (actually, this is the main justification of studying a broad taxonomic sample). One bad consequence of quick studies is that they in practice discourage subsequent careful work, even by the same group. This brings the second major problem I mentioned: although the findings of the authors suggest that Y chromosomes from distant Diptera families evolve very similarly to the pattern well documented in *Drosophila*, this similarity is basically not acknowledged in the paper, as even a cursory read of the Conclusions demonstrate. For these omissions I see no justification within the scope of good science.

We would like to point out that we cited 8 papers (citation nr. 7, 8, 9, 10, 11, 12, 17, 18; i.e. over 20% of our original citations) that describe the gene content of the *Drosophila melanogaster* Y chromosome (from B. Carvalho and colleagues) in our original manuscript throughout the paper, in both the introduction, and the results & discussion section, but not in our brief conclusion. However, we now also include citations to these papers in the conclusion section, and include even more references to previous work in other systems.

Thus, in my opinion there are important methodological problems in the paper which would prevent its acceptance in its present form. I also think that the presentation and discussion of the results is misleading at several passages, either by presenting as new well known results obtained other groups, or by failing to openly acknowledge the limitations of the methods and analysis employed. I detailed below these problems, in the same order of the five main findings presented above. An unrelated problem is that I could not find in the Nat. Comm. website the "Supplementary file 1", which is essential for fully reviewing the paper (it contains the sequence of the genes found in the Y chromosome of all species). I ask the Editor to double-check, may be I am missing something.

1. Computational pipeline.

Methodological issue: The pipeline has many arbitrary parameters. How were they chosen? In particular, was *D. melanogaster* used to fine-tune the pipeline parameters? If yes, then it is not appropriate to use this species to evaluate the pipeline efficiency, because it amounts to a sort of over-fitting. This information should be included in the Methods section.

Our pipeline was not fine-tuned to fit *D. melanogaster*. Instead, as stated in our manuscript, it was mainly modeled after a previous study in mammals by Cortez et al. Nature 2014, keeping in mind (a) the relatively low coverage we had for RNA-seq and gDNA-seq; and (b) the rather fragmented nature of most of our genome assemblies. In fact, our pipeline appears to do better for some other species, where running it in reverse directions fails to identify any female-specific transcripts (in contrast to *D. melanogaster*, where we identify three female-specific transcripts). The accuracy of our pipeline is confirmed by showing similarly high stringency in other fly species that have well-assembled genomes. As stated above (response to point 5 of Reviewer 1), we manage to identify both Y-linked genes in *Anopheles* that were found to be Y-linked and present in multiple strains by Hall et al. 2016 (and we detected no female-specific transcripts in *A. gambiae*).

In addition, we now have also verified our pipeline in additional species with well-assembled genomes (*Anopheles gambiae* & *D. pseudoobscura*), and find that the vast majority of putative Y-linked genes identified by our pipeline map to either Y-linked scaffolds in these species (3 out

of 3 in *Anopheles gambiae*), or unmapped scaffolds in the assembly (which are presumably Y-linked; this is the case for 11 out of 12 putative Y-linked transcripts in *D. pseudoobscura*; and 59 out of 63 putative Y-linked transcripts in *D. miranda* map either to the neo-sex chromosomes, or unplaced scaffolds).

Finally, we now also include male-specific PCR amplification of many putative Y-linked genes in several non-Drosophila fly taxa for which we had male and female samples available (a total of 24 putative Y-linked transcripts were validated by PCR in *Themira minor*; *Teleopsis dalmanni*; *Ephydra hians* and *Phortica variegata*). We discuss this more clearly now on page 7 and Table S4.

Data versus conclusions: In several paragraphs the authors stated that "Overall, our pipeline shows both high sensitivity and specificity for detecting Y-linked genes, especially for species and genomic regions with high read coverage". However, in 9 out of 24 species it did not yield any candidate, or yield too many non-sensical results (as defined by the authors, "too many" means more than 1/3 of the transcripts matching the XX female genome but not XY males; Table S5). Even among the 15 species that were analysed, there is a fair number of "female-specific" transcripts which are false positives, since female DNA is a subset of the male DNA. For example, in *E. hians* (which has a good transcriptome dataset), the authors reported 28 male specific and 9 female specific transcripts. Hence, we would expect that among these 28 transcripts (deemed as Y-linked), 9 would be false-positives. Failure in 9 out of 24 species, and an estimated rate of false positives of 9 out of 28 transcripts are not what most people will understand by high sensitivity and specificity; it is more likely an exploratory tool. There is no major problem on using exploratory tools, unless this is not openly acknowledged.

We have greatly expanded our discussion on why in some species our pipeline is less successful in identifying Y-linked sequences (see also response point 1 to Reviewer 1). As stated above, we fail to identify Y-linked transcripts in species that have homomorphic sex chromosomes or are XO (such as the Hessian fly), and thus simply lack male-specific sequences. We also have difficulties in reliably identifying male-specific genes in species with very large inferred genomes (i.e. >500Mb), where our genome assemblies tend to be highly fragmented. However, most of our species have genome sizes considerably smaller, and our pipeline works well inferring male-specific sequences in this case. As stated above, we now have also verified our pipeline in additional species with well-assembled genomes (*Anopheles gambiae* & *D. pseudoobscura*), and by male-specific PCR amplification of many putative Y-linked genes in several non-Drosophila fly taxa. This is all included in the manuscript on page 6-7. *E. hians* was by far the worst species in our original data set (with potentially 9 false positives out of 28 transcripts); however, all the other species fared substantially better when comparing the true and false positive (see Table S3). *E. hians* had good transcriptome data but a very large and poorly assembled genome (see point above that our pipeline performs worse on larger genomes). However, we have now included additional genomic sequencing as well as RNA-seq data for this species (see Table S1), and now this species performs substantially better as well: we now infer 9 male-specific (i.e. putative Y-linked transcripts), and 2 female-specific transcripts. We openly state in our paper that our method is not perfect and will identify some false-positive transcripts (it even does so in our benchmark species *D. melanogaster*); however, the fraction of false-positives we estimate is 1.9% (16 female-specific genes in 859 sex-specific contigs identified in total), and many of the non-Drosophila putative Y-genes were confirmed experimentally). Thus, this relatively low number of false positive will not affect any of the conclusions in our paper.

Acknowledgement of previous work : Other groups studying Diptera Y (mosquitoes and Drosophila) developed specific methods to identify Y-linked genes, which were published in major genomic journals, and that do not seem to suffer from the above mentioned problems. None were discussed (or cited) in the paper.

Six novel Y chromosome genes in Anopheles mosquitoes discovered by independently sequencing males and females

Hall et al BMC Genomics. 2013; 14: 273.

Carvalho, A.B., and A.G. Clark. Efficient identification of Y chromosome sequences in the human and *Drosophila* genomes. *Genome Research*, 23: 1894-1907, 2013.

For the benefit of the readers, it would be desirable that the authors discuss/compare what they propose to what is available in the literature.

We did try both the Carvalho & Clark method and the Hall et al. method to identify Y-linked candidate genes, but they both did not work for our species (i.e. they produced way too many false positive in *D. melanogaster* and other flies). Briefly, the difference is that the two methods were developed to identify Y-linked genes in flies that have already high-quality genome assemblies (such as *D. melanogaster* – the C&C method; or *Anopheles* – the Hall et al. method). In particular, by comparing male and female sequence data to a reference genome, Y-linked sequences can be identified based on being present only in the male sequence data (either by identifying scaffolds with male-specific kmers or by finding scaffolds with higher read coverage in male than female genomic reads). Our initial application of these approaches to our male and female genomic fly data was of limited success to reliably identify Y genes, presumably due to a combination of factors: Y chromosomes have few genes and mainly consist of repetitive DNA, and our genome assemblies from next-generation sequencing data are more fragmented, especially at repeat-rich regions. Thus, fragmented genome assemblies combined with moderate genomic read coverage prevented us from using methods to infer Y-linked genes simply based on genomic data. This is now explicitly mentioned in the paper on page 4.

2. Diptera Y-linked genes were aquired from the autosomes, instead as being relics from the original autosome pair that originated the sex-chromosomes (as happens in mammals).

Acknowledgement of previous work: the pattern found by the authors actually has been thoroughly documented in *Drosophila*, starting in 2000 (reviewed in Carvalho et al TIG 2009). For example, Carvalho Lazzaro and Clark (PNAS 2000) described their findings as follows: " A striking pattern emerges from the phylogeny of the Y dyneins: they all are closely related to other *Drosophila* genes, but none of these paralogous genes is X-linked (Fig. 4). The same pattern occurs with PRY.". Thus, the pattern initially discovered in *Drosophila* seems to be valid for Diptera in general. This similarity is at most tersely acknowledged in the ms, or not acknowledged at all, as a cursory reading of the Conclusion section shows: Diptera and mammalian Y are compared, and external references are provided only for the mammalian work, which would induce readers to believe that this is the first paper that investigate Y-chromosome evolution in Diptera. Analogous omissions are presented in many places of the ms. What the authors really found is that Y evolution of other Diptera families seem very similar to previously documented in *Drosophila*; this should be openly acknowledged. Mosquito data is more scanty and less clear, but should also be discussed in this context.

The Carvalho Lazzaro and Clark (PNAS 2000) was cited in our original submission (reference 9); and we also cited 7 additional papers on *D. melanogaster* Y-linked genes (citation nr. 7, 8, 10, 11, 12, 17, 18; i.e. over 20% of our original citations) from B. Carvalho, A. Clark and colleagues in our original manuscript throughout the paper, in both the introduction, and the results & discussion section, but not in our brief conclusion. In addition, we cited several neo-sex chromosome papers from *Drosophila* (20, 21, 22, 23) as well as a mosquito paper (reference 19) studying the evolution of Y-linked genes. We now also include citations to these papers in the conclusion section, and include even more references to previous work in other systems, throughout the manuscript.

3. The acquisition of Y-linked genes occurred gradually (Fig 4).

Acknowledgement of previous work: Similar to #2 above. This pattern was well documented in *Drosophila*. See, for example, Fig 2 of Koerich et al Nature 2008. and Fig. 4 of Carvalho and Clark Genome Research 2013.

Methodological issues: Ka Ks analysis between the Y-linked gene and a paralog was used to roughly estimate the time of duplication that generated the Y-linked genes. Hence it is essential that the choosen paralog is the correct one, which is not trivial and, I think, not possible to be done with the uncurated computational approach they used. The

authors should provide a table linking each Y-linked gene to its paralog, including *D. melanogaster*. The fasta sequence of these paralogs should also be provided, since many genomes were not previously annotated.

The Koerich et al Nature 2008 paper was cited in the original manuscript, and we did mention in the Introduction that Y-linked genes were “*acquired secondarily on the Y, after it evolved its male-limited transmission*” – and we cited references 9-12; i.e. four of Carvalho & colleagues’ papers, including the Koerich et al Nature 2008 paper. We now also add citations to this paper and others at the specific paragraph where we discuss gradual acquisition of Y-linked genes, and explicitly mention that patterns of gradual gene acquisitions found in Diptera are consistent with previous results in *Drosophila*.

The referee is right that it is difficult to unambiguously identify paralogs, and we put a lot of effort trying to correctly identify paralogs for our putative Y-linked transcripts. As stated in the method, we used stringent parameters to infer paralogs: We require at least 40% identity, a blast bit score of at least 50 and in case of several blast hits, we only take the one where the e-value is at least an order of magnitude lower than the second best hit; transcripts with several blast hits with similar e-values were not included in our analyses. Our analysis appears stringent in identifying paralogs: in *D. pseudoobscura*, for example, the 30 genes for which we identified paralogs, 26 mapped to previously annotated genes and the remaining 4 map to genomic regions that have expression tracks based on flybase, but are not annotated as genes. We also include the sequences / fasta files of paralogs in our revision as File S2..

4. This genes already have male-related functions before being acquired by the autosomes (Fig 5, part).

Acknowledgement of previous work: Similar to #2 and #3 above. This pattern was well documented in *Drosophila* since 2000:

Carvalho and Clark Genome Research 2013: " Hence, these four genes fit the general pattern of *Drosophila* Y-linked genes, formerly autosomal male-specific genes transposed to the Y-chromosome (Carvalho et al. 2009)." Again, no clue of these previous results in the ms.

Again, we do cite these observations and several of Carvalho’s and Clark’s paper in the introduction “*Intriguingly, most Y-linked genes in Drosophila are not simply remnants of genes present on the autosome that became the sex chromosome; instead, they all appear to have been acquired secondarily on the Y, after it evolved its male-limited transmission [9-12]. Y-linked genes in D. melanogaster all have male-specific functions and have adapted testis-specific expression, which suggests that they were acquired from autosomes and retained on the male-specific Y because of male-beneficial functions*” and in the results “*Previous Work [12, 19] and our analysis suggests that most genes on ancestral Y chromosomes were acquired from autosomal locations.*” However, we now include additional citations to these observations by Carvalho and Clark where we first cite Figure 6A, and also in the conclusions.

Table S5: Please add a column with the sex-chromosome system (XY ZW , or homomorphic). This will help the readers to understand the results.

We added a column (now Table S3).

"(though the application of long-read PacBio technology has provided useful in assembling individual Y-linked genes in *D. melanogaster*;" Actually PacBio allowed the assembly of fairly large genomic regions of the Y, containing several genes / pseudogenes .

We modified that statement.

Reviewers' comments:

Reviewer #1 (Remarks to the Author):

This manuscript largely confirms what previously has been found with respect to patterns of Y chromosome evolution in *Drosophila melanogaster*, but also shows that these patterns seem to hold across a range of other dipteran species. The study is thus of great interest, as it shows that the pattern found for *D. melanogaster* is general and not evolutionary oddity. Presentation of results has improved substantially in the revised version, especially with respect to the figures that previously were in a poor shape. I just have a few minor comments I'd like to authors to address/look over.

Fig 4. It looks like the X-axis has changed, and that the "X" transcripts are less centered over -1. Why is this, is this correct? I am not sure about adding the background shading, there must be many autosomal and X linked transcripts that are found in the "wrong" shading area. Adding a histogram for *D. miranda* makes it easier to see how the Y linked transcripts are distributed, but I think readers need to be informed of why there are two panels for this species.

Fig S1 Please spell out in the figure legend, as indicated in the response letter, that multiple blue lines represent multiple isoforms?

Fig S5 The Y-axis needs to be explained in the legend.

Reviewer #2 (Remarks to the Author):

Overall, the manuscript has been significantly improved. Most notably, the figures are much clearer, and include much more relevant info that was previously omitted. Also, the authors have made extensive changes throughout the manuscript that definitely improve its clarity. As a result the current version is almost an entirely new manuscript, and due to the limited amount of time, I was unable to fully go over all changes. Nonetheless, I am convinced the authors have substantially improved on their previous version.

Relevant to my previous comments, I am content with the manner in which the authors have addressed my questions and criticism. Most importantly, the following changes address my main critiques of this manuscript. The authors argue that the conservation of synteny has been shown in a previous study, as is indeed the case. The inclusion of this information is adequate, as it suggests their claim that these genes moved onto the Y-chromosome after it became a Y-chromosome, and not before it became one (on p9). Also, the authors now discuss gene expression of Y-linked genes and the role thereon of male-limited selection more clearly, emphasizing that genes may be preferably Y-linked due to having a male-specific function or having a male-specific function due to Y-linkage.

Although the manuscript has been improved considerably, some issues remain that require attention. Most of these are not major issues, and therefore would not require a great deal

of work. However, I would like to see some clarification for some of the changes made by the authors (see Major comments).

Major comments:

P6-7: Here, the authors discuss verification of Y-linkage by PCR (which definitely strengthens the manuscript). Please include the generated data as supplementary data (e.g. gel pictures), along with the relevant information on how these data were generated (DNA extraction, PCR protocols, etc.).

Compared to the original manuscript, some of the numbers have changed, and I am not always sure as to why this is done, and more importantly which of these numbers are correct. Perhaps the authors could go over all the numbers in the manuscript to verify them.

- P3: Authors mention 22 species (rather than 24), though the abstract mentions 15? And I believe ultimately only 13 are used (e.g. as seen in fig. 3)? Is this because Y-linked genes were not identified in 9 species, and 2 species with ZW chromosomes were omitted in the revised version? (this is discussed on p6 of the revised manuscript).

- On p. 7, the authors mention:

"In total, we identified 187 protein-coding transcripts (or parts of transcripts), and 656 non-coding transcripts across all species that are potentially Y-linked."

In the original manuscript, these numbers are 184 and 650 respectively (p5 of original MS). How did this difference come to be (perhaps I am missing something in the rebuttal?)

- Also p7: *D. miranda* now has 59 potentially protein-coding sequences rather than 56?

- P13/14: almost all the numbers in the section on *D. miranda* have changed?

Minor comments:

P20: Table S5 should be Table S3 (now that the tables are fused)?

Figure 1: 'Larvae' and 'pupae' are in plural, but 'embryo' and 'adult' in singular forms; use one for all.

Figure 5: Y-axis for Ka/Ks values seems cut off wrong compared to original figure (though it is greatly improved in terms of resolution).

Some minor spelling errors occur throughout the manuscript, consider having a fresh pair of eyes read your manuscript to fix these. For example, there are some inconsistencies/errors with species' names:

P10: Authors use both '*A. gambiae*' and '*An. gambiae*' to refer to *Anopheles gambiae*.

P13: '*pseuoobscura*' should be '*pseudoobscura*'

Table S3: '*Sarcophaga bulata*' should be '*Sarcophaga bullata*' (still needs to be changed).

Response to Reviewer #1:

This manuscript largely confirms what previously has been found with respect to patterns of Y chromosome evolution in *Drosophila melanogaster*, but also shows that these patterns seem to hold across a range of other dipteran species. The study is thus of great interest, as it shows that the pattern found for *D. melanogaster* is general and not evolutionary oddity. Presentation of results has improved substantially in the revised version, especially with respect to the figures that previously were in a poor shape. I just have a few minor comments I'd like to authors to address/look over.

Fig 4. It looks like the X-axis has changed, and that the "X" transcripts are less centered over -1. Why is this, is this correct? I am not sure about adding the background shading, there must be many autosomal and X linked transcripts that are found in the "wrong" shading area. Adding a histogram for *D. miranda* makes it easier to see how the Y linked transcripts are distributed, but I think readers need to be informed of why there are two panels for this species.

Thank you for pointing this out – we made an error in rescaling the X-axis (natural log instead of log2) and that has been corrected; the X-linked transcripts are now centered over -1 and autosomal ones at 0. We kept the shading as it makes it visually easier to gauge whether paralogs of Y-linked transcripts are X-linked or autosomal, though the reviewer is correct that some transcripts may be found in the wrong shading area. We spell that out in the Figure legend. We also spell out why we provide a histogram for *D. miranda* in the Figure legend.

Fig S1 Please spell out in the figure legend, as indicated in the response letter, that multiple blue lines represent multiple isoforms?

This is done now.

Fig S5 The Y-axis needs to be explained in the legend.

We now explain the Y-axis in the Figure legend.

Response to Reviewer #2:

Overall, the manuscript has been significantly improved. Most notably, the figures are much clearer, and include much more relevant info that was previously omitted. Also, the authors have made extensive changes throughout the manuscript that definitely improve its clarity. As a result the current version is almost an entirely new manuscript, and due to the limited amount of time, I was unable to fully go over all changes. Nonetheless, I am convinced the authors have substantially improved on their previous version.

Relevant to my previous comments, I am content with the manner in which the authors have addressed my questions and criticism. Most importantly, the following changes address my main critiques of this manuscript. The authors argue that the conservation of synteny has been shown in a previous study, as is indeed the case. The inclusion of this information is adequate, as it suggests their claim that these genes moved onto the Y-chromosome after it became a Y-chromosome, and not before it became one (on p9).

Also, the authors now discuss gene expression of Y-linked genes and the role thereon of male-limited selection more clearly, emphasizing that genes may be preferably Y-linked due to having a male-specific function or having a male-specific function due to Y-linkage.

Although the manuscript has been improved considerably, some issues remain that require attention. Most of these are not major issues, and therefore would not require a great deal of work. However, I would like to see some clarification for some of the changes made by the authors (see Major comments).

Major comments:

P6-7: Here, the authors discuss verification of Y-linkage by PCR (which definitely strengthens the manuscript). Please include the generated data as supplementary data (e.g. gel pictures), along with the relevant information on how these data were generated (DNA extraction, PCR protocols, etc.).

We now include that information in the Methods section, and we provide PCR pictures as a supplementary figure (Fig. S4).

Compared to the original manuscript, some of the numbers have changed, and I am not always sure as to why this is done, and more importantly which of these numbers are correct. Perhaps the authors could go over all the numbers in the manuscript to verify them.

Some of the numbers changed since we include more data for a few species (as mentioned in our previous response), and we also slightly modified our cut-off for detecting homologs and coding transcripts. We double-checked each number to make sure they are correct.

- P3: Authors mention 22 species (rather than 24), though the abstract mentions 15? And I believe ultimately only 13 are used (e.g. as seen in fig. 3)? Is this because Y-linked genes were not identified in 9 species, and 2 species with ZW chromosomes were omitted in the revised version? (this is discussed on p6 of the revised manuscript).

Yes, the reviewer is correct. We investigate 22 species in total (we excluded the two ZW species from our initial submission that contained 24 species), and exclude an additional 9 species for which we could not identify any Y-linked transcripts. This is discussed in detail on p. 6, and we also change the abstract to state that we investigated 22 species in total.

- On p. 7, the authors mention:

"In total, we identified 187 protein-coding transcripts (or parts of transcripts), and 656 non-coding transcripts across all species that are potentially Y-linked."

In the original manuscript, these numbers are 184 and 650 respectively (p5 of original MS). How did this difference come to be (perhaps I am missing something in the rebuttal?)

As mentioned above, some of the numbers changed since we include more data for a few species (as mentioned in our previous response), and we also slightly modified our cut-off for detecting homologs and coding transcripts. We double-checked each number to make sure they are correct.

- Also p7: *D. miranda* now has 59 potentially protein-coding sequences rather than 56?

- P13/14: almost all the numbers in the section on *D. miranda* have changed?

See above. We double-checked each number to make sure they are correct. We realized that we were using the default blat threshold in *D. miranda* (instead of the cutoff that we were using for other species) in order to identify Y-linked transcripts, which added 30 additional transcripts to the final *D. miranda* list.

Minor comments:

P20: Table S5 should be Table S3 (now that the tables are fused)?

Changed.

Figure 1: 'Larvae' and 'pupae' are in plural, but 'embryo' and 'adult' in singular forms; use one for all.

We have fixed the use of pupa and larva in Fig 2 and Fig 6.

Figure 5: Y-axis for Ka/Ks values seems cut off wrong compared to original figure (though it is greatly improved in terms of resolution).

We changed the axis of the Ka/Ks Figure (and plot all Ka/Ks values > 3 at 3), in order to have better resolution for Ka/Ks values for all the other species. This is now clearly stated in the legend.

Some minor spelling errors occur throughout the manuscript, consider having a fresh pair of eyes read your manuscript to fix these. For example, there are some inconsistencies/errors with species' names:

P10: Authors use both 'A. gambiae' and 'An. gambiae' to refer to *Anopheles gambiae*.

P13: 'pseuoobscura' should be 'pseudoobscura'

Table S3: 'Sarcophaga bulata' should be 'Sarcophaga bullata' (still needs to be changed).

Thanks for pointing this out. We have gone through our manuscript and double-checked the spellings.